

# Warming increases soil respiration in a carbon-rich soil without changing microbial respiratory potential

Marion Nyberg[1], Mark J. Hovenden[1]

[1]School of Natural Sciences, University of Tasmania, Hobart, 7001, Australia

*Correspondence to:* Marion Nyberg (current affiliation University of British Columbia) (mnybe1@mail.ubc.ca)

**Abstract.** Increases in global temperatures due to climate change threaten to tip the balance between carbon (C) fluxes, liberating large amounts of C from soils. Evidence of warming-induced increases in $CO_2$ efflux from soils has led to suggestions that this response of soil respiration ($R_S$) will trigger a positive land C–climate feedback

cycle, ultimately warming the earth further. Currently, there is little consensus about the mechanisms driving the warming-induced $R_S$ response, and there are relatively few studies from ecosystems with large soil C stores. Here, we investigate the impacts of experimental warming on $R_S$ in the C-rich soils of a Tasmanian grassy sedgeland, and whether alterations of plant community composition or differences in microbial respiratory potential could contribute to any effects. *In situ*, warming increased $R_S$ on average by 28% and this effect was consistent over

time and across plant community composition treatments. In contrast, warming had no impact on microbial respiration in incubation experiments. Plant community composition manipulations did not influence $R_S$ or the $R_S$ response to warming. Processes driving the $R_S$ response in this experiment were, therefore, not due plant community effects and are more likely due to increases in belowground autotrophic respiration and the supply of labile substrate through rhizodeposition and root exudates. $CO_2$ efflux from this high-C soil increased by more

than a quarter in response to warming, suggesting inputs need to increase by at least this amount if soil C stocks are to be maintained. These results indicate the need for comprehensive investigations of both C inputs and losses from C-rich soils if efforts to model net ecosystem C exchange of these crucial, C-dense systems are to be successful.

**1 Introduction**

Globally, more carbon (C) is stored in soils than the amount of C in the atmosphere and in plants combined (Canadell et al., 2007). Simple physiology suggests that soil respiration ($R_S$) rates will increase as soil temperatures rise (Gillooly et al., 2001), stimulating $CO_2$ emissions from the soil – a response that has the potential to outweigh plant productivity responses to global warming and lead to a net loss of C from soils (Melillo et al.,

2017). Recently, numerous studies have suggested that global warming is indeed disturbing the balance between ecosystem C inputs and outputs (Melillo et al., 2017). This suggests the possibility of a positive feedback whereby warming increases C efflux from soils, which accelerates climate change leading to further C losses and so on (Bridgham et al., 2008; Melillo et al., 2017; Bond-Lamberty et al., 2018). Importantly, it is possible that warming-induced C losses increase with soil C content such that is has been suggested that soils storing the most C could

shift from C sinks to C sources (Cai et al., 2010; Wilson et al., 2016; Jassey et al., 2018).

Increases in respiration of soil organic carbon (SOC) as an effect of experimental warming occur almost universally (Rustad et al., 2001) however, increasing soil temperatures stimulate not only soil microbes and



enzyme activity, but also net primary productivity (NPP) and fresh C input from litterfall, and root exudations (Rustad et al., 2001), enhancing substrate availability for microbial respiration (Lu et al., 2013; Wang et al., 2017).

Warming effects have also been demonstrated to drive microbial priming, whereby decomposition is enhanced through increased input of labile C compounds (van der Wal and de Boer, 2017). Despite this, greater above ground plant biomass is not directly linked to immediate or long-term increases in the storage of SOC and hence the mechanisms driving the response of Rs to warming are uncertain (Jackson et al., 2017).

Effects of temperature on environmental factors such as soil moisture, substrate availability and
evapotranspiration also influence and mediate rates of decomposition of SOM, and efflux of $CO_2$ (Davidson et al., 2000; Eliasson et al., 2005; Lu et al., 2013). These effects include extension of growing seasons and shifts in species composition and community structure (Chen et al., 2016). Considering this, changes in plant community composition, and subsequent shifts in functional traits have the potential to influence the quantity and quality of organic matter in the soil, as well as the physical soil structure (Metcalfe et al., 2011). This suggests that there is
potential for the response of soil C dynamics to warming to be partially or even wholly dependent upon changes to plant community composition (Jackson et al., 2017).

Both experimental and global warming have impacts on soil water availability, which is itself a primary determinant of Rs (Schimel et al., 1994). Following a unimodal relationship, respiration is highest at an intermediate (35-50% by volume) soil water content (SWC), which stimulates microbial activity and enhances
above and below ground labile C inputs (Chou et al., 2008; Zhou et al., 2010; Wang et al., 2017). Anaerobic conditions in wet and flooded soils suppress microbial activity, slowing decomposition of SOM (Davidson and Janssens, 2006). Similarly, low SWC can have a similar effect by reducing microbial activity, restricting soil respiration (Carey et al., 2016). As warming generally leads to lower soil water content (Zhang et al., 2013; Li et al., 2017), the impact on Rs depends upon the underlying soil water content, increasing respiration of wet soils
but reducing respiration in drier soils (Almagro et al., 2009). Essentially, the effect of warming on SWC could either offset or exacerbate direct warming effects on soil respiration, potentially disturbing the entire global C balance.

Substrate availability is another factor that is affected by warming, and thus has the potential to shift the temperature sensitivity of SOM decomposition (Davidson and Janssens, 2006). Largely, increased temperatures
lead to the loss of physical or chemical protection of SOM, and thus enhanced microbial respiration of soil organic carbon (SOC) (Davidson and Janssens, 2006). Partitioning SOC into pools as a function of recalcitrance and residence time assists with analysing effects of environmental manipulations on long-term C storage (Pendall et al., 2011). As C inputs to the soil and consequently into these various pools occur in response to the interplay between rates of NPP, decomposition, climatic conditions and soil characteristics (Ontl and Schulte, 2012), the
fate of SOC is either transformation into highly recalcitrant humus, important for the stabilisation and long-term storage of SOC, or loss to the atmosphere as $CO_2$. Thus, factors such as oxygen availability, substrate quantity and quality, nutrient limitation and activity of extracellular enzymes are key to the soil respiration response. Carefully controlled laboratory incubations are necessary to eliminate confounding factors and pinpoint the mechanisms driving responses observed in the field (Davidson and Janssens, 2006). Ultimately, distinguishing
between the potential driving factors is vital for our ability to model future C fluxes and to extend the observations from field experiments more widely.



To understand the consequences of warming on soil C dynamics and particularly Rs, it is necessary to distinguish between warming-related increases in Rs that are simply due to an increase in the biochemical response of Rs to temperature, and potential alterations of the temperature sensitivity of Rs caused by climate warming. The increase
in Rs with rising temperature has been widely documented (Luo et al., 2001; Rustad et al., 2001), however, the temperature sensitivity of Rs in soils that have undergone experimental warming is much more variable (Song et al., 2014; Carey et al., 2016). Shifts in the temperature sensitivity under warming are likely to be driven by both changes in microbial community composition and changes in the physical and chemical properties of the soil (Davidson and Janssens, 2006). Additionally, effects of warming such as soil drying affect various ecosystem
processes and thus might shift the temperature response of Rs (Carey et al., 2016; Moinet et al., 2018). The effect of temperature on Rs is thus complex, and there are a number of biotic and abiotic factors influencing the response of SOM decomposition to warming. Until these various influences are characterised accurately, projecting future soil C emissions will remain problematic.

Although measurements of soil respiration *in situ* often demonstrate warming-related increases, the mechanisms
behind this response cannot be revealed by simple field observations (Davidson and Janssens, 2006). In particular, it is difficult to distinguish changes in microbial community composition and functioning in response to warming from *in situ* measurements alone. These changes include acclimation (Luo et al., 2001) or adaptation (Bradford et al., 2008), encompassing both physiological and genetic changes within individuals and species, changes in community structure (Sheik et al., 2011) and a shift towards microbial use of slowly decomposing C (Bracho et
al., 2016). Hence, a shift in temperature sensitivity of SOM decomposition is likely to be driven by warming through a change in microbial respiratory potential, expressed as the $CO_2$ mineralisation rate.

Here, we use a manipulative experiment to examine the potential influences of climate change, specifically warming and plant community composition, on soil C dynamics. We examine soil respiration responses both *in situ* and in laboratory incubation experiments to disentangle the mechanisms involved in the response of soil
respiration to both warming and manipulation of the plant community. Specifically, we ask the following questions:

     1.) Does warming increase soil respiration in a Tasmanian C-rich soil?

     2.) If so, is this due to changes in microbial respiratory potential?

     3.) Does altering plant community composition change the response of soil respiration to warming?


## 2 Materials and Methods

### 2.1 Study site

All field measurements and soil samples were taken at the Silver Plains warming experiment in the Tasmanian central plateau, Australia (42°09'S, 147°08'E; 890 m a.s.l). The site is a natural grassy sedgeland with an average
summer temperature of 16°C, average winter temperature of 6°C and average annual rainfall of 720 mm (BOM, 2018). Soil at the site is peaty, being an organosol containing on average 8 kg C m-2 in the top 10 cm. The vegetation at the site is heavily grazed year-round by a range of native vertebrate herbivores, including wallabies,



pademelon and wombats, as well as by feral fallow deer, resulting in an extremely low vegetation stature of a few cm, with the exception of inflorescences which can extend up to 30 cm above the ground.


### 2.2 Experimental design

The experiment was set up in the 2014 austral winter as a fully orthogonal, two-factor random block design, with warming and species removal as fixed factors across eight replicate blocks (Fig. 1). The experiment consists of forty 2 x 2 metre plots, with 3 metres between each plot, of which 20 were warmed using hexagonal polycarbonate

open-top chambers (OTC) with an internal diameter of 1.5 m, with the remainder of being unwarmed, ambient plots. To investigate the impact of altering plant community composition, the dominant species, *Poa gunnii*, was removed by plucking in one warmed and one ambient plot (henceforth "dominant removal" plots) in each block. One warmed and one ambient plot in each block was left untouched (henceforth "no removal" plots). To control for possible effects of removing biomass during the dominant species removal treatment, we also removed

biomass from one additional warmed and unwarmed plot in every second block, removing the same amount of biomass as was removed from the "dominant removal" plots in the same block but with this biomass removed randomly from across the plot, rather than from a single species (henceforth "random removal" plots). Plant biomass was removed in the spring and summer of 2014/15 by gently removing small plants by hand and by repeatedly clipping larger plants to ground level until green shoots no longer emerged. The amount of biomass

removed in each plot is presented in Table S1. After the initial removal treatment, all plots were left undisturbed until the following spring, at which time all plots were surveyed to determine whether removed plants had re-established. As removed plants had not re-established at this time, no further removal occurred. Plant biomass was not measured directly in the plots in order to reduce disturbance. However, measures of vegetation cover and height indicated that the vegetation in removed plots had recovered completely within two years and were very

similar to untouched control plots by this time, except in terms of species composition.





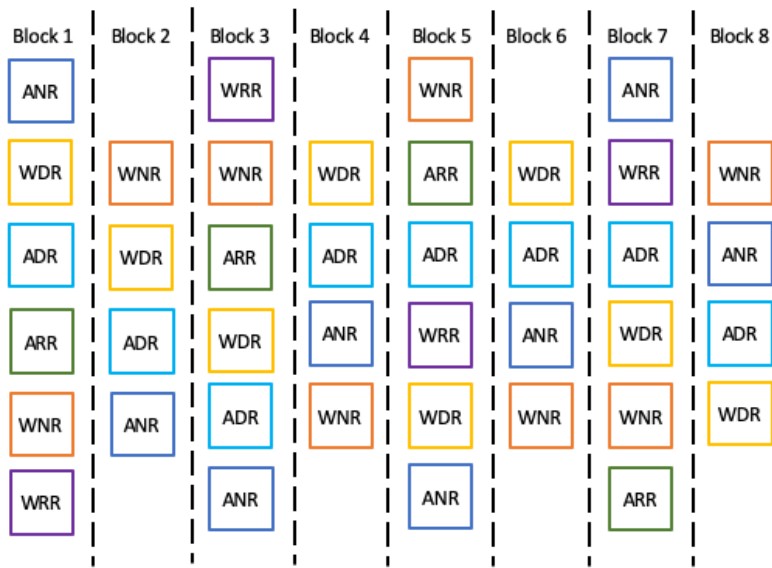

**Figure 1. Conceptual diagram for the experimental design of the Silver Plains warming experiment. Each block contains a warmed and unwarmed plot with no species removed (WNR) and (ANR) respectively; a warmed and unwarmed plot with the dominant species removed (WDR) and (ADR) respectively; and in every second block, i.e. in**

**four blocks, there is a warmed and unwarmed plot with random biomass removal (WRR) and (ARR) respectively.**

Air temperature at 5 cm height and soil temperature at 5 cm depth in each plot was logged continuously with iButton dataloggers. Over the entire five year period, the warming treatment increased air temperature 5 cm above the soil surface by 1.56°C ($P<0.004$) and soil temperature at 5 cm depth by 1.29°C ($P<0.001$).

**2.3 *In situ* methods**

A 50 mm length of 100 mm diameter PVC pipe was inserted into the soil to a depth of 2 cm, extending 3 cm above ground height, within the centre 0.25 m2 of each plot for soil respiration measurements. Soil respiration was measured with a $CO_2$/$H_2O$ infrared gas analyser (IRGA) (Licor, model LI-6400) with attachment of a Licor 6400-09 soil chamber, which attached to PVC pipes. Soil respiration was measured *in situ* monthly from August

2017 to June 2018. On each occasion, three measurements of *in situ* soil respiration, defined as the $CO_2$ efflux rate, were made in each plot. The average value of these three measurements was used in subsequent analyses. Soil temperature and moisture in each plot were measured at the exact same time as the soil respiration measurements on each occasion. Soil temperature was measured with a soil thermocouple probe (LiCor 6000-09TC) attached to the LI-6400. Volumetric soil water content (SWC) was estimated at 5 locations in each plot

using a hand held TDR probe at 0-5cm depth. The 5 separate measurements of SWC where then averaged to obtain one SWC value per plot on each measuring occasion.





Six randomly placed soil samples, amounting to a total of approximately 25-30 g fresh weight, were collected from each plot using a 1.5 cm diameter hand corer to a depth of 5 cm below ground level, twice throughout the year. Samples were collected on the 02/03/18, representing the end of summer, or growing season soil, and on the
25/06/18, representing winter soils.

### 2.4 Laboratory incubations

Soil cores collected *in situ* were immediately placed on ice for return to the laboratory, where they were refrigerated (4°C) overnight. The following day, the samples were composited at the plot level and sieved through
a 4 mm sieve for one minute to remove leaves and large roots. A 10 g fresh-weight sub-sample was removed and oven dried from each composite sample for the determination of total soil C. Each subsample was ground to a powder in a Retsch Mixer Mill (MM200, Retsch GmbH, Haan) and then C content was analysed by combustion in a Perkin Elmer 2400 Series II Elemental Analyser (Perkin Elmer Australia, Melbourne). The remaining soil was used immediately for laboratory incubations to determine microbial respiration, as detailed below.

Microbial respiration as a function of temperature was determined by incubation using soils sampled in the Silver Plains warming experiment at the end of summer and in mid-winter 2018. For each plot, three replicate samples weighing four to eight grams from the composite sample were placed in 100 mL specimen jars, each of which was incubated at a different temperature. Each sample was wetted to bring them to 90% of field capacity for winter soils and 60% of field capacity for summer soils to represent prevailing soil moisture conditions in each
respective season. Once water was added to all soil samples, specimen jars were placed in 500 ml preserving jars with tightly fitting lids containing a septum to allow gas headspace samples to be collected by syringe. Jars were stored in dark incubation cabinets at temperatures at one of 10, 17 or 25°C, with one sample from each plot at each temperature. Headspace gas of jars were sampled (20 ml) using a syringe on days 1, 2, 4, 5, 7, 9, 12, 15, 19, 23, 29, 35, 49, 56, 63. After extracting samples from each jar, headspace samples were analysed for $CO_2$
concentration, representing soil respiration, and microbial respiratory potential was thus defined as the rate of $CO_2$ release. To analyse headspace gas, samples were injected directly into an infrared gas analyser (LI-6262, Li-Cor, Lincoln, NE). After measurements were taken and analysed, jars were ventilated for 20 minutes and headspace gas equilibrated with atmospheric air. Following this, lids were replaced and headspace gas was sampled and analysed again to obtain starting $CO_2$ concentration for each jar. C mineralisation over the sample period was
calculated from the increase in headspace $CO_2$ concentration.

Total C mineralisation over the entire incubation period was simply the sum of the amount of C mineralised over each sample period. Daily C mineralization results (dC/dt) were analysed using non-linear curve fitting routines in R (version 3.4.3, R Core Team, 2017), with a single pool plus constant model (Pendall et al., 2011) to estimate the size of the labile C pool ($C_a$), the intrinsic decay constant of the labile pool ($k$), and the intrinsic decay constant
of the stable C pool ($Y_0$):

$$\frac{dC}{dT} = C_a k e^{-kt} + Y_0$$

(1)





### 2.5 Data analysis

Field soil respiration rates were analysed using a 2-factor repeated measures ANOVA with warming and removal as the fixed factors. Since soil temperature ($T_S$) and SWC are known controllers of $R_S$ and varied substantially over the year, we also analysed field $R_S$ with a 2-factor ANCOVA with $T_S$ and SWC and the interaction between $T_S$ and SWC as covariates. Treatment means were calculated as least-squares means using the lsmeans package to account for the influences of covariates (Russel V. Lenth, 2016). Treatment effects on SWC and $T_S$ were

analysed using 2-factor repeated measures ANOVA exactly as for $R_S$.

Because there was a significant influence of warming on $R_S$, we created a separate model of the influence of SWC and $T_S$ on *in situ* $R_S$ for warmed and unwarmed plots. Since the respiration temperature relationship is best described by an Arrhenius-type function (Fang & Moncrieff 2001), we used multiple regression techniques to fit an exponential relationship to $R_S$ and SWC, $T_S$ and the interaction between $T_S$ and SWC. Such a non-linear

relationship fitted the observed data far better than a linear model, as compared by the Akaike information criterion corrected for finite sample size.

Total cumulative $CO_2$ emitted in laboratory incubations, $C_a$, $k$, and $Y_0$ for each season were compared using three-factor analysis of variance ANOVA for both summer and winter soils with incubation temperature, warming and species removal as fixed factors, including all interactions. Seasonal differences were also analysed using four-

factor ANOVA, with season also included as a fixed factor along with warming effect, removal and incubation temperature.

All statistical analyses were carried out in R (version 3.4.3). Data were checked for heteroscedasticity and normality and the required transformations were made using the Box Cox power and logarithmic transformations. Significant treatment effects were further analysed using Tukey's HSD *post hoc* comparisons.


### 3 Results

### 3.1 *In situ* soil respiration

### 3.1.1 CO$_2$ efflux

Experimental warming drove a significant increase in soil respiration over the course of the year ($F_{1,12}=58.48$,

$P<0.001$; Table 1) but there was no significant influence of the species removal treatment, so neither the dominant nor random removal treatments were different to the untouched plots ($F_{2,12}=1.1$, $P=0.36$), nor was there a warming x removal interaction effect on $CO_2$ efflux ($F_{2,12}=0.14$, $P=0.87$). As expected, time of year had a strong effect of $CO_2$ efflux ($F_{6,12}=11.84$, $P<0.001$), with the highest rates, $13.23 \pm 0.37$ μmol $CO_2$ m$^{-2}$ s$^{-1}$, in summer, decreasing through to $1.4 \pm 0.06$ μmol $CO_2$ m$^{-2}$ s$^{-1}$ in winter (Table 1). Despite the strong variation in C efflux rates across

the year, there was no significant interaction between month and warming ($F_{5,12}=1.17$, $P=0.38$), indicating that the warming effect was consistent across the year.



**Table 1. The impact of experimental warming on soil $CO_2$ efflux, soil temperature and soil water content in the Silver Plains Warming Experiment from August 2017 to June 2018. Values shown are means with standard errors in parentheses (n=20).**

| Month | Treatment | $CO_2$ efflux $\mu mol\ CO_2\ m^{-2}\ s^{-1}$ | Soil temperature °C | SWC % |
|---|---|---|---|---|
| August | Ambient | 1.8 (0.1) | 5.1 (0.1) | 34.3 (0.6) |
| | Warmed | 2.3 (0.1) | 5.3 (0.1) | 27.4 (0.5) |
| November | Ambient | 8.0 (0.3) | 16.0 (0.3) | 19.8 (1.0) |
| | Warmed | 12.2 (0.4) | 16.9 (0.3) | 16.5 (0.7) |
| January | Ambient | 11.6 (0.3) | 17.1 (0.2) | 19.3 (0.4) |
| | Warmed | 13.2 (0.4) | 17.8 (0.2) | 17.5 (0.6) |
| February | Ambient | 7.6 (0.2) | 18.9 (0.1) | 13.3 (0.2) |
| | Warmed | 12.9 (0.5) | 18.8 (0.1) | 12.6 (0.3) |
| March | Ambient | 6.1 (0.1) | 13.3 (0.3) | 20.6 (0.3) |
| | Warmed | 9.0 (0.2) | 13.5 (0.2) | 15.8 (0.3) |
| April | Ambient | 4.7 (0.1) | 12.6 (0.2) | 13.7 (0.3) |
| | Warmed | 7.3 (0.1) | 13.2 (0.1) | 11.1 (0.3) |
| May | Ambient | 4.0 (0.1) | 10.9 (0.3) | 1225 (0.3) |
| | Warmed | 5.7 (0.1) | 12.3 (0.2) | 9.8 (0.2) |
| June | Ambient | 1.4 (0.1) | 3.3 (0.2) | 52.0 (1.7) |
| | Warmed | 1.7 (0.1) | 4.1 (0.1) | 45.3 (1.2) |


### 3.1.2 Soil temperature





Time of year had a strong impact on soil temperature ($F_{6,12}$=27.61, P<0.001), which varied from 3.33 ± 0.18 °C

to 18.89 ± 0.14°C over the study period. Experimental warming had a significant impact on soil temperature

increasing soil temperature at 5 cm depth by 0.55°C on average ($F_{1,12}$=7.31, P=0.02). This impact was sustained

over the year with no significant month x warming interaction ($F_{5,12}$=0.88, P=0.52), indicating that the warming

chambers had a similar effect on soil temperature across the year.  Neither removal treatment, i.e. neither dominant

nor random biomass removal ($F_{2,12}$=1.99, P=0.18), nor warming x removal interactions ($F_{2,12}$=0.45, P=0.65)

affected soil temperature. Thus, the warming treatment increased soil temperatures consistently over the year and

across the species removal treatments.

### 3.1.3 Soil water content

Soil water content (SWC) also varied over the year ($F_{6,12}$=6.21, P=0.003) reflecting precipitation patterns at Silver

Plains (BOM, 2018). Over the course of the year, SWC ranged from 9.83 ± 0.17% to 52 ± 1.69%, with moisture

levels decreasing from winter 2017 through to autumn 2018, and then steeply increasing again in winter 2018

(Table 1). Experimental warming significantly decreased SWC throughout the year by 3% on average (P<0.001),

which is expected considering the drying effect of warming. However, the impact of warming on SWC depended

upon the month, as indicated by a significant sampling month x warming effect ($F_{5,12}$=6.09, P=0.005). Warming

had the greatest effect on SWC in August 2017 and June 2018, when SWC was highest and soil temperature was

lowest. SWC in these winter months was substantially higher than during the rest of the year, with SWC on

average 34 and 58% higher in August and June respectively, than the overall mean value (21.3 ± 0.5%). The

proportional reduction in SWC due to warming in these months was nearly two-times the yearly average.

Otherwise, the warming effect was similar between sampling months. There was no significant influence of the

removal treatment, i.e. neither the dominant nor random removal treatments were different to the untouched plots

($F_{2,12}$=0.23, P=0.8), nor was there a warming x removal interaction effect on SWC ($F_{2,12}$=0.52, P=0.61), again

indicating that plant species removal did not alter the influence of the warming treatment.

### 3.1.4 Relationships between environmental factors and $CO_2$ efflux

Both soil temperature ($F_{1,33}$= 33.62, P<0.001) and SWC ($F_{1,33}$= 5.95, P=0.02) were strong controllers of soil $CO_2$

efflux over the year at Silver Plains (Fig. 2). However, treatment effects on these abiotic factors alone were

insufficient to explain the higher C efflux in warmed plots, as ANCOVA indicated that the warming treatment

still induced significant increases in $CO_2$ efflux when variation in soil T and SWC were accounted for ($F_{1,33}$=

44.83, P<0.001). Thus, the warming treatment increased soil $CO_2$ efflux independently of its effects on soil

temperature and SWC (Fig. 2). Across the whole year LS mean $CO_2$ efflux rates for ambient soils was 6.07

(C.I=5.69,6.45) µmol $CO_2$ m$_{-2}$ s$_{-1}$ but 8.48 µmol $CO_2$ m$_{-2}$ s$_{-1}$ (CI=8.09,8.86) for warmed soils, amounting to a

warming-induced increase of 28% at a common soil temperature and SWC. As $CO_2$ efflux measurements spanned

a large variation in both soil T and SWC, it was possible to discern a trend whereby the stimulation of C efflux

by warming became more pronounced as soil temperature increased (Fig. 2). Neither removal, i.e. neither

dominant nor random biomass removal($F_{2,33}$=0.89, P=0.42), nor a warming x removal interaction ($F_{2,33}$=0.57,

P=0.57) affected $CO_2$ efflux, as indicated by ANCOVA.

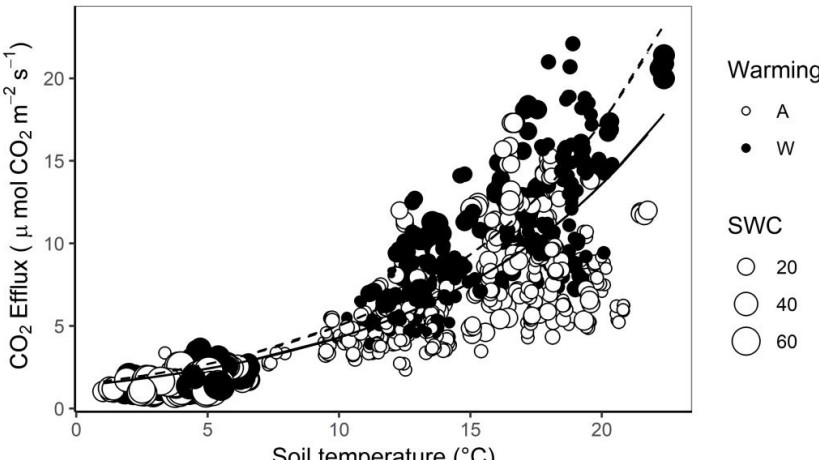

**Figure 2. CO₂ efflux as a function of soil temperature and soil water content for warmed (W) and ambient (A) plots at Silver Plains from August 2017 to June 2018. Size of the point represents SWC %, with larger points corresponding to higher SWC. The regression lines indicate the relationship between CO₂ efflux and soil temperature at median SWC**
**in ambient plots (solid line) and warmed plots (dashed line).**

### 3.1.5 Models of CO₂ efflux

As ANCOVA indicated that soil CO₂ efflux at Silver Plains was significantly influenced by soil temperature, SWC and a strong warming effect, the relationship between these covariates and CO₂ efflux could be estimated
separately for ambient and warmed treatments. First a general regression model of CO₂ efflux was fit and selected using model selection based on AICc. The most parsimonious and accurate model was one which included soil temperature ($T_S$), SWC, and a SWC x $T_S$ interaction term (Int.term).

This model was then fit independently to ambient and warmed plots using the relative coefficient values, with 89% of the variance in CO₂ efflux explained in warmed plots Eq. (2) and 82% in ambient plots Eq. (3).

$$CO_2 \text{ efflux}_{ambient} = e^{(-0.8+0.359\,\log(SWC)+0.115\,(T_S)+0.003(Int.term))}$$

$$R^2 = 0.82$$

$$(2)$$

$$CO_2 \text{ efflux}_{warmed} = e^{(-0.06+0.148\,\log(SWC)+0.124\,(T_S)+0.002(Int.term))}$$

$$R^2 = 0.89$$

$$(3)$$

Thus, it is possible to model CO₂ efflux across a range of soil temperature and SWC values in both ambient, unwarmed (Fig. 3A) and warmed conditions (Fig. 3B). From these plots, it is possible to determine that while the CO₂ efflux rate increases more steeply with rising temperature in warmed plots than in unwarmed plots, the way


in which it does so is also dependent upon the SWC (Fig. 3A and B). Thus, the impact of experimental warming

on soil $CO_2$ efflux was greatest in warm ($T_s > 15°C$) relatively dry conditions (SWC<30%; Fig. 4).

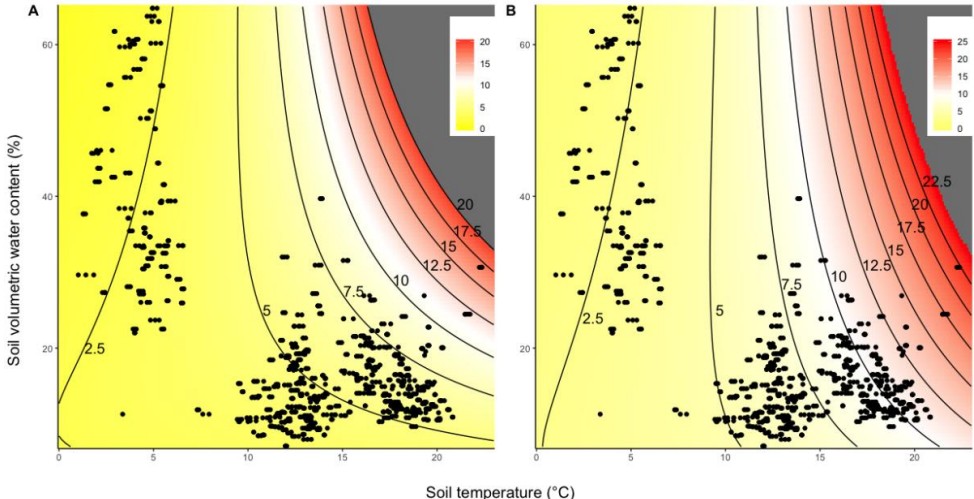

**Figure 3. $CO_2$ efflux ($\mu$mol $CO_2$ m-2 s-1) modelled as a function of soil temperature and SWC in A.) ambient plots and B.) warmed plots. Colour indicates predicted $CO_2$ efflux values and field observations are shown as individual points. Regions beyond the observed range of $CO_2$ efflux rates are shown in grey.**



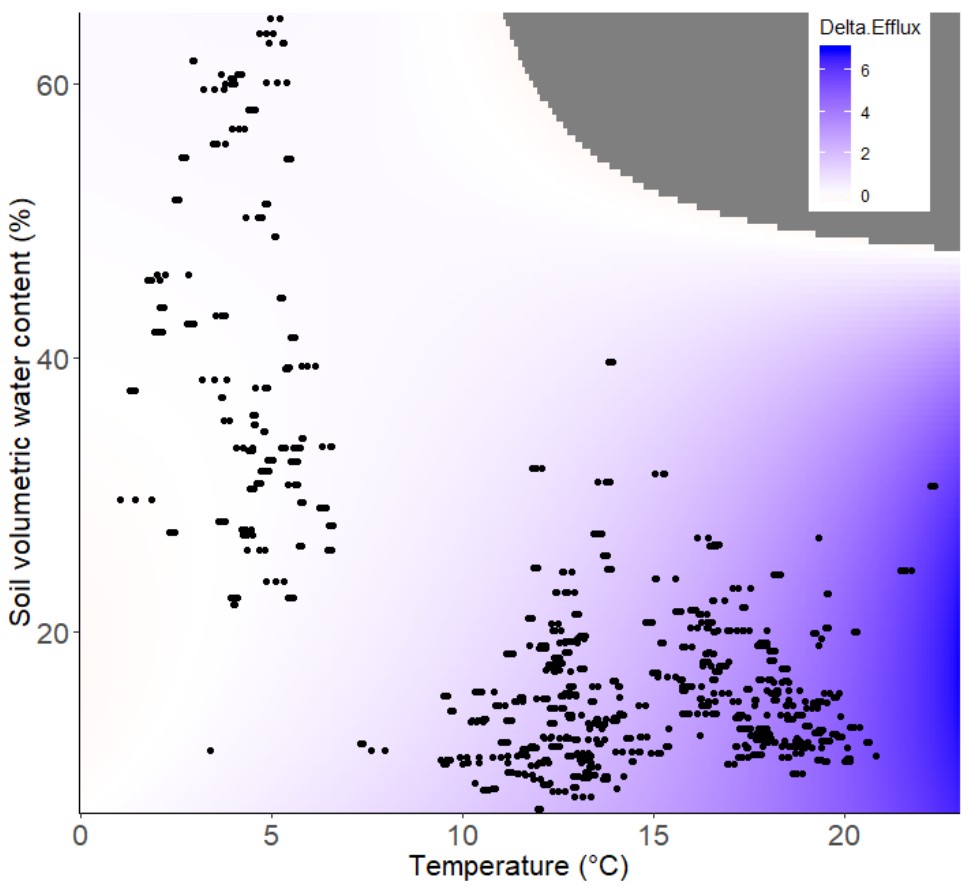


**Figure 4. Delta $CO_2$ efflux ($\mu$ mol $CO_2$ m-2 s-1). The amount of extra $CO_2$ that is likely to be released due to warming as a function of soil temperature (°C) and SWC (%). Data points represent actual measurements and colour indicates predicted $CO_2$ efflux. Points on the contour graphs are field observations and regions beyond field observations, and thus where $CO_2$ cannot be predicted, are greyed out.**


### 3.2 Laboratory incubations

### 3.2.1 Total C mineralisation

To determine whether experimental treatments altered potential microbial respiration, soil samples were collected in summer and winter for laboratory incubations. These incubations allowed the temperature sensitivity of soil respiration, the size of the labile C pool ($C_a$) and its decay constant ($k$) to be assessed, as well as estimating the decay constant of the more resilient stable C pool ($Y_0$) to be assessed in constant, optimal conditions. From soils collected in summer, the total amount of C mineralised increased substantially as an effect of incubation temperature, however there were no effects of either the warming or removal treatments. On average, soil incubated at 17°C for two months emitted 48% more C than at 10°C, and a further 22% at 25°C ($F_{2,82}=80.9$,





P=<0.001; Fig. 5). From soils collected in winter, total C mineralised again only increased significantly as an effect of incubation temperature, with on average a 26% increase in C emitted at 17°C from 10°C, and a further 27% increase at 25°C ($F_{2,112}$=49.56, P<0.001; Fig. 5). Just like the situation with soil collected in summer, there were no treatment effects on the total amount of C mineralised from winter soils ($F_{1,112}$=0.04, P=0.84). Between seasons, winter soils emitted on average 24% less C than summer soils ($F_{1,196}$=33.66, P<0.001), most likely because of the higher SWC used for the winter soils, and neither removal treatment, i.e. neither dominant nor random biomass removal ($F_{2,196}$=0.67, P=0.51), nor warming significantly affected total C mineralised overall ($F_{1,196}$=0.01, P=0.92).

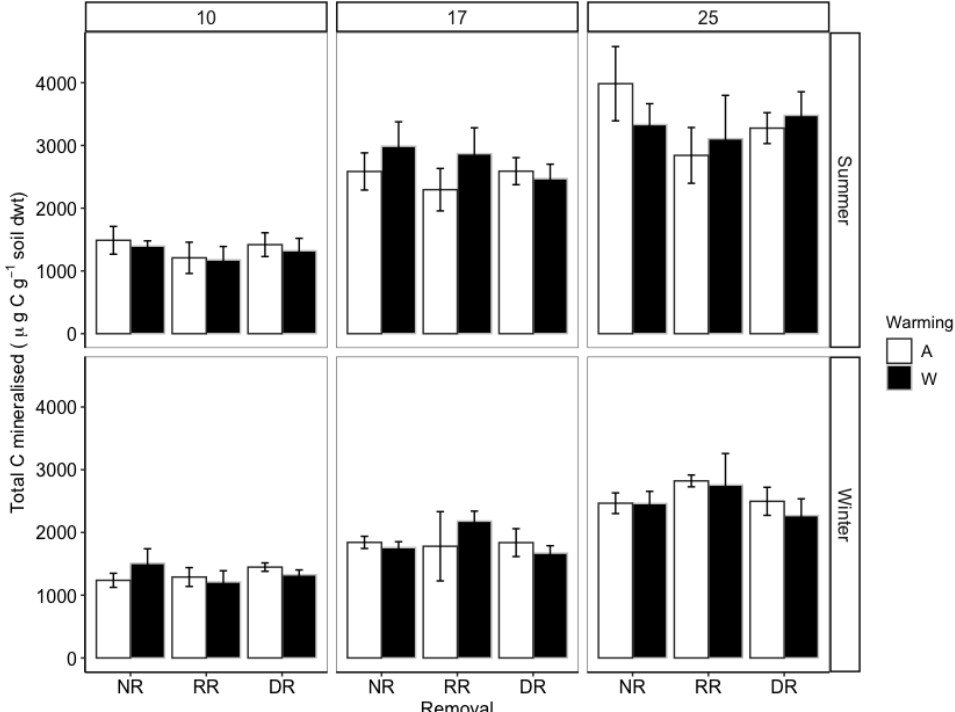

**Figure 5. Total C mineralised in summer and winter from soils in no removal (NR), (random removal (RR), and dominant removal (DR) plots at incubation temperatures, 10, 17 and 25°C for warmed (W) and ambient (A) treatments.**

### 3.2.2 Labile C

In summer soil, incubation temperature significantly increased the size of $C_a$ on average by 50% from 10°C to 17°C, and by a further 18% at 25°C (P<0.001) (Fig. 6A). There were no treatment effects on the size of $C_a$. Winter soil incubations reflect similar results to those for summer soils, with a 27% increase in $C_a$ pool size from 10°C to 17°C, and a further 27% increase to 25°C (P=0.001). As with summer soil there were no treatment effects. Overall, season had no effect on $C_a$, however incubation temperature increased $C_a$ across the two seasons of 36% from 10°C to 17°C and a further 24% at 25°C (P<0.001).





The intrinsic decay constant of the labile pool ($k$) in summer soil was not affected by incubation temperature

($F_{2,82}$=0.39, P=0.68), the warming ($F_{1,82}$=0.06, P=0.8) , or removal treatments, i.e. neither dominant nor random biomass removal ($F_{2,82}$=0.31, P=0.73), was significantly influenced by an interaction between warming and species removal ($F_{2,82}$=3.14, P=0.05) (Fig. 6C). In ambient plots, removing the dominant species tended to increase $k$, however, in warmed plots, the opposite occurred. Post hoc analysis revealed the greatest differences in $k$ were observed between warmed x no removal and warmed x dominant removal plots, and warmed x dominant

removal and ambient x dominant removal plots. In winter, there were no treatment or incubation temperature effects on $k$, however $k$ was on average 42% greater in summer ($F_{1,196}$=201.09, P<0.001).

### 3.2.3 Intrinsic decay constant of the stable C pool

From summer soil, the size of the stable C pool ($Y_0$) also increased significantly ($F_{2,82}$=78.01, P<2-16) as a function

of incubation temperature with an average increase of 47% from 10°C to 17°C, and a further 20% at 25°C (Fig. 6C). There were no treatment effects on the $Y_0$ of summer soil. For winter soils, responses to treatments were similar to those of summer soils. There were no treatment effects, but incubation temperature increased $Y_0$ on average by 27% from 10°C to 17°C, and a further 28% at 25°C ($F_{2,112}$=45.9, P<0). Overall $Y_0$ was 39% higher in summer than in winter ($F_{1,196}$=137.61, P<0.001), and incubation temperature also significantly increased $Y_0$

overall, with on average a 38% increase from 10°C to 17°C, and a further 23% at 25 °C ($F_{1,196}$=107.28, P<0.001), however there were no treatment effects.

### 3.2.4 Proportion of total C that was labile

From summer soil, the proportion of total C that was from $C_a$ was only affected by incubation temperature with

on average a 49% increase from 10°C to 17°C, and a further 22% increase when incubated at 25°C ($F_{2,82}$=77.73, P<0.001; Fig. 6D). There were no treatment effects. Similarly, in winter, the proportion of total C that was $C_a$ increased only as a function of increasing incubation temperature, with on average a 24% increase from 10°C to 17°C and a further 27% at 25°C ($F_{2,112}$=22.19, P<0.001). Overall, the proportion of total C that was $C_a$, increased substantially as a function of incubation temperature ($F_{2,196}$=67.94, P<0.001) with a 35% increase from 10°C to

17°C, and a further 25% increase at 25°C, however there were no overall treatment effects.


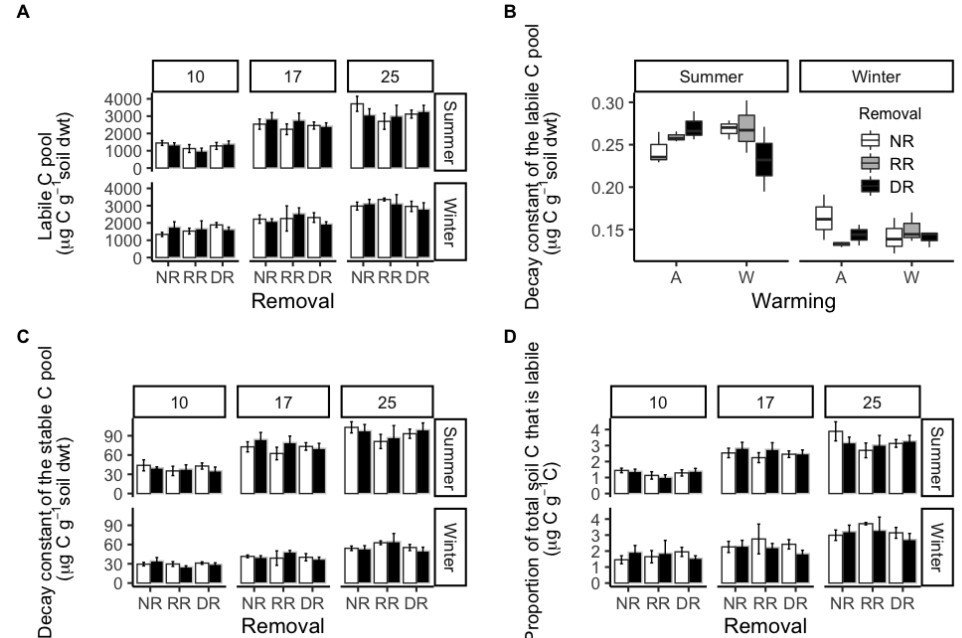

**Figure 6.** A.) Labile C pool size (C$_a$) in summer and winter soils in no removal (NR), random removal (RR) and dominant removal (DR) plots at incubation temperature, 10, 17 and 25°C for warmed (*black*) and ambient (*white*) treatments. B.) Intrinsic decay constant of the labile C pool (k) both summer and winter soils in no removal (NR), random removal (RR), and dominant removal (DR) plots at incubation temperature, 10, 17 and 25°C for warmed (*black*) and ambient (*white*) treatments. C.) The intrinsic decay constant of the stable C pool (Y$_0$) in summer and winter soils in no removal (NR), random removal (RR), and dominant removal (DR) plots at incubation temperature, 10, 17 and 25°C for warmed (black) and ambient (white) treatments. D.) Proportion of total C that is from the labile C pool (C$_a$) in both summer and winter soils in no removal (NR), random removal (RR), and dominant removal (DR) plots at incubation temperatures 10, 17 and 25°C for warmed (black) and ambient (white) treatments.

### 3.3 Total soil C content

Overall, irrespective of removal treatment, total soil C % averaged 19.2 ± 0.4 (P<0.001). C % was 18.7 ± 0.7 and 19.7 ± 0.6 in ambient and warmed soils respectively, however there were no significant treatment effects .

All incubation results were also analysed per gram of soil C but results were essentially identical to those expressed per gram of soil dry weight, above.

### 4 Discussion

The main aim of this study was to investigate whether warming increases R$_s$ *in situ*, and whether any observed treatment effects were due to an increased ability of the soil microbial community to mineralise SOC. Additionally, we investigated whether manipulating plant community composition affected the R$_s$ response to warming. Results





demonstrated strong warming-related increases in Rs *in situ*, however there were no warming effects on microbial respiratory potential. Additionally, the warming treatment increased soil temperature and decreased soil water
content significantly *in situ*, however the warming effect on Rs was greater than expected considering the impact it had on these abiotic factors. Thus, warming increased Rs more than simply by increasing soil temperature and reducing soil water content. Manipulating the plant community by removing the dominant species or removing biomass had no impact on Rs, nor did these treatments influence the impact of warming on Rs. This suggests that the warming-induced increase in Rs was independent of any influences on plant community composition. Similarly,
removal treatments did not affect microbial respiratory potential, however there was a complex warming and removal interaction that influenced the decay constant of the soil labile C pool ($k$). Overall, the results from this study suggest that as there was no change in microbial respiratory potential, the observed increase in soil respiration *in situ* was largely an effect of altered plant activity in warmed plots.

### 395 4.1 Possible mechanisms leading to the warming-induced increase in soil respiration

Warming increased Rs *in situ* over the course of the sampling period from November 2017 to June 2018. This increase in $CO_2$ efflux observed from soils *in situ* encompasses the response of both microbial (heterotrophic) respiration, and plant root (autotrophic) respiration, and amounted to an average increase in soil C efflux of 28%. The observed increase of Rs in response to warming is in line with multiple other studies, although most of these
focus on soils in the low- to mid-range of soil C stocks and in northern hemisphere locations (Lu et al., 2013; van Gestel et al., 2018). There are 4 possible mechanisms whereby could have increased in Rs: 1.) Increased temperature sensitivity of Rs; 2.) Influence through change in plant community composition; 3.) Enhanced substrate supply through SOM and 4.) Plant induced alteration to soil microhabitat. The substantial Rs response to warming could be due to one or a combination of these processes and determining which were likely to be
involved has significant ramifications for our ability to predict future soil C dynamics.

#### 4.1.1 Increased temperature sensitivity of Rs

One of the proposed mechanisms behind the increased Rs response to warming, and subsequent loss of soil C stores is an increase in the temperature sensitivity of Rs, i.e. increased decomposition of SOM (Kirschbaum,
1995). This response, mainly attributed to an increase in enzyme kinetics with temperature, is linked strongly to substrate availability (Davidson and Janssens, 2006). At Silver Plains, the overall significant increase in Rs rates from warmed plots *in situ* implied that the temperature sensitivity of Rs was higher under warming. The highest Rs rates were recorded during the growing season in spring and summer, suggesting primary productivity, microbial activity and environmental factors such as precipitation are likely to substantially influence respiration
rates (Almagro et al., 2009). However, despite the strong dependence of Rs on soil water content and soil temperature, warmed plots had higher rates of C efflux from the soil under particular combinations of soil temperature and moisture (Section 3.1.5, Fig. 3). The restrictive effect of high soil water content and low soil temperature on Rs observed in this study is widely documented and due to the creation of anoxic conditions limiting microbial access to substrate (Schimel et al., 1994; Syed et al., 2006; Sierra et al., 2015). Hence the
observed effect of soil water content and soil temperature on Rs was anticipated, however the degree to which





warming enhanced the response of Rs to temperature, was greater than expected. This observation could be explained by the greater effect of warming on air temperature than soil temperature at 5cm, thus considering most microbial activity occurs in the surface soil layers, it is possible the soil was warmed more than the amount measured, partially accounting for the large increase in Rs. Interestingly, the degree of stimulation by warming increased as soil temperature increased, i.e., there was a greater warming-induced stimulation of C efflux when soil was warm than when it was cold. This contrasts previous findings that indicate a greater warming effect on Rs at lower temperatures (Wang et al., 2014), and results from the incubation experiment in this study that reveal warming did not affect microbial respiration at any incubation temperature.

The large apparent increase in Rs observed *in situ* implied that warming possibly triggered an increase in microbial respiratory potential. Partitioning of Rs in incubation experiments allows the response of microbial respiration to warming to be observed under optimal conditions, controlling for soil water content and soil temperature. In stark contrast to the *in situ* observations, soil incubations revealed no differences in the temperature sensitivity of microbial respiration between warmed and ambient soil. There were no differences among treatments in the total C mineralisation rate measured under laboratory conditions, which would indicate that the ability of the soil microbial community to mineralise soil C was unchanged. This lack of any treatment effect was similar in winter and summer-collected soils, even though there was a strong seasonal effect on the $CO_2$ respiration rate in incubated soils. Winter soils emitted significantly less $CO_2$ than soils collected at the end of summer, a response attributed to decreased access to substrate as an effect of limited enzyme activity in cold temperatures (Suseela et al., 2012). Additionally, despite claims that the warming-induced increase in Rs is due to a strong, positive relationship between the average turnover time of labile C pools and mean annual air temperature (Trumbore et al., 1996), there was no difference in the size of the respired labile C pool ($C_a$) between warmed and ambient soils, or between seasons. Therefore, this suggests that the warming treatment did not increase the temperature sensitivity of labile C decomposition. Additionally, there were no warming or removal treatment effects on the decay constant of the stable C pool ($Y_0$), calculated from $CO_2$ emission rates late in the incubation period. This implies that stable C, which is chemically and physically protected (Schlesinger, 1997), was not sensitive to warming, a response that contrasts results obtained elsewhere (Leifeld and Fuhrer, 2005; Hartley and Ineson, 2008). Thus, lack of a warming effect indicates that warming-induced increases in labile, or stable C temperature sensitivity are not driving the Rs response to warming observed *in situ*.

Essentially, the incubation studies revealed that four years of experimental warming had not altered either the potential for microbial respiration or its inherent temperature response, as soils incubated at the same temperature respired more or less at the same rate, regardless of whether they were collected from warmed or ambient plots. These results indicate that the warming-induced stimulation of Rs *in situ* was not due to changes in the inherent temperature response of microbial respiration. Considering the soil incubation experiment decoupled microbial respiration from plant activity and particularly C inputs, the lack of a warming treatment effect on C emissions in the incubation experiments is evidence that plants play a large role in the respiration response.

**4.1.3 Influence through change in plant species community composition**



Plant community compositional change drives ecosystem responses to global changes, particularly when it involves shifts in the dominance or abundance of plant functional types (Bret-Harte et al., 2008). This is particularly true with Rs and global warming, as warming-related changes in plant functional types, and hence the resources they input to the soil, are highly likely to occur (Saleska et al., 2002). By investigating the effects of dominant species removal, random biomass removal, and warming on the response of respiration, there is scope to gain insight into future ecosystem dynamics under a changing climate. Removal of a dominant species from an ecosystem has promoted species diversity and altered ecosystem function, implying dominants reduce the establishment of other species (Wardle et al., 1999). Metcalfe et al., (2011) highlight the significant role that functional traits of the dominant species hold on many soil processes, including decomposition and respiration and hence one would expect to observe these effects in this study. Plants modify local soil conditions through root exudations of hormones, sugars, phenolics and amino acids, essentially structuring the rhizosphere microbial community composition. This means that changes in plant community composition have the potential to affect Rs and thus ecosystem functioning (Van Nuland et al., 2016), and hence the motive to investigate how Rs responds to the combination of warming and manipulated plant community composition. Results from Silver Plains demonstrate neither removal of the dominant plant species nor random removal of biomass had any effect on Rs or the temperature response of Rs *in situ*, and very little effect on microbial respiratory potential. This suggests that, in line with previous studies, temperature had greater control on Rs than variation in plant community composition (Duval and Radu, 2018). This result is surprising because despite removing the dominant species, there did not appear to be any functional shifts within the community, suggesting it may have been replaced by a functionally-similar species, or there was a compensatory response by functionally-different species (Bret-Harte et al., 2008). Differences in plant chemistry, morphology and physiology affect the quantity and quality of root and leaf litter, leading to changes in SOM decomposition rates, shifts in microbial respiratory potential and community structure (Van Nuland et al., 2016). However, the response of Rs *in situ* indicates that despite removal of the dominant species or random removal of biomass, community function was maintained.

Despite the absence of an effect on Rs *in situ* from manipulating species composition, microbial respiration dynamics indicated that warming and biomass removal (both random and dominant removal treatments) reduced the intrinsic decay constant of the labile C pool ($k$) in summer soil. Interestingly, biomass removal had the opposite effect in ambient plots. In a previous clipping experiment, which is representative of biomass removal, a decrease in Rs due to clipping was explained as relocation of assimilates to shoots, reduction in the supply of photosynthates to roots, and thus decreased root respiration (Zhou et al., 2010). Hence there is likely to be less available substrate under warming and biomass removal scenarios, and $k$ is therefore lower. Considering this, the interactive effect of warming and biomass removal on $k$ is complex and requires further investigation to explore the mechanistic basis behind the response. The absence of an influence on Rs through variation in plant community composition suggests this mechanism is not driving the warming-induced Rs response to warming.

### 4.1.4 Enhanced substrate supply

**SOM**



As SOM forms mainly from plant litter, warming related increases in both above and belowground primary productivity suggest supply of SOM will be greater under warming (Rustad et al., 2001; Lin et al., 2010; Wu et al., 2011), at least in systems that are not water limited. Additionally, experimental warming often increases leaf-drop, root-turnover and the subsequent decomposition of leaf and root litter (Lu et al., 2013), with the combined effects of warming and higher C inputs on respiration rates reported to be greater than the impact of either factor

in isolation (Hopkins et al., 2014). Root and leaf litter have fast turnover times, implying they represent a major source of C for microbial decomposition. Therefore, an increase in the input of easily degradable C would promote microbial activity (Wan et al., 2005; Hogberg and Read, 2006), stimulating soil C efflux. Considering this, an increase in substrate supply seems like a conceivable explanation for the increase in Rs observed *in situ*. However, incubation experiments indicated no influence of warming on the total amount of C between warmed and ambient

plots, or on the size of the labile pool ($C_a$), or total C respired. This indicates that substrate supply and availability from plant biomass is similar in warmed and ambient plots. Previous investigations suggest that despite warming-related increases in litter quantity, enhanced respiration due to increased labile C concentration in soils is likely to offset additional C inputs (Lu et al., 2013), meaning changes to both inputs and losses of soil C could balance each other. Interestingly neither *in situ* Rs, nor total soil C or $C_a$ was affected by plant community composition

manipulations, suggesting substrate supply and availability was similar regardless of warming and removal treatments. This result contrasts with those from previous clipping experiments that demonstrated that biomass removal limits substrate supply (Wan and Luo, 2003; Xue et al., 2016). Overall this suggests that increased substrate supply through SOM is not a driving mechanism behind the warming-induced increase in Rs observed *in situ* although specific tests of this mechanism, such as through the use of stable isotope tracing, would be

required to be confident.

### 4.1.6 Plant induced alteration to soil microhabitat

The final mechanism that could be driving the warming-induced increases in Rs are plant induced alterations to the soil microhabitat. In this study, as in most, soil for incubations was not analysed as intact soil cores, rather

being sieved and homogenised, altering the microhabitat conditions. This is potentially problematic, as it is assuming that rhizosphere processes are not influencing the overall Rs response. Previous studies have demonstrated the important role roots play in stabilising SOM (Hinsinger et al., 2009), with disturbed soils having a lower capacity to protect SOM due to mechanical disruption of macroaggregates, and hence C is more readily decomposed by microorganisms (Beare et al., 1994). Additionally, macrofauna such as earthworms and

nematodes play an important role in the early stages of SOM decomposition (Wardle et al., 2004), and therefore the absence of these species from the incubations could have influenced the rates of C efflux. Considering this, it is possible that through homogenisation of soil in incubation studies, soil C dynamics and decomposition rates are confounded by disturbances to the soil microhabitat.


### 4.3 Implications



Large C stocks within this type of peaty habitat imply they are globally important for the C cycle, thus understanding potential losses are immensely important for the global C budget. Previous studies on the response of Rs to warming have been largely centred around northern hemisphere sites, and with that there has been large unexplained variability in the response (van Gestel et al., 2018). This implies that the mechanisms behind the Rs response to warming are poorly characterised. Our results indicate that warming-related increases in $CO_2$ efflux from C-rich soils in grassy peatlands are expected in the future, however microbial respiratory potential is not the driving factor, and thus there is a strong link to plant activity and C inputs. Moreover, the results indicate that the impact of warming on soil $CO_2$ efflux is strongly dependent upon both soil temperature and moisture conditions, improving the confidence that current and future soil $CO_2$ efflux can be modelled from these variables. However, the lack of observations in certain combinations of soil temperature and moisture mean that predictions using the models presented here should be limited to the observed range. Future work should test the generality of these models in previously unobserved combinations of soil moisture and temperature.

Predictions regarding future climate conditions require a more comprehensive mechanistic understanding of temperature and decomposition relationships, especially considering the global variation in these relationships. Further investigation into the role of inputs is required, as warming could be driving increases in inputs, thereby balancing the accelerated C efflux and preventing net loss of C from soils. Alternatively, warming could lead to depletion of huge stores of C. This effect is no doubt subject to great variation depending on the ecosystem and hence the necessity to examine the response, accounting for heterogeneity in soil and vegetation types worldwide. Most importantly, this study revealed that C inputs through root exudates and root respiration were the two mechanisms most likely to be driving the Rs response to warming. Thus, more research into the influence of root exudates and root respiration on Rs, particularly under warming, will provide a more comprehensive insight to the Rs response. Ultimately, thorough investigations into the whole ecosystem C exchange is required to advance understanding into how warming will affect rates of inputs and outputs.

The increase in Rs in response to warming observed here is in line with previous experimental warming studies, although few have been conducted in C-rich soils. Thus, the results from this study contribute directly to a field of knowledge that is currently underrepresented. Despite a strong warming effect, there appear to be no significant effects of plant community manipulation, suggesting that warming exerts more of an influence on $CO_2$ efflux from soils than differences in plant communties. Additionally, results suggest that the microbial respiratory potential in this system is not altered by experimental warming and hence cannot be decoupled from plant activity if we are to enhance our ability to predict C cycling dynamics in a warmer climate. Current findings suggest warming is likely to trigger a positive feedback cycle whereby increases in global temperatures will enhance $CO_2$ efflux from soils, subsequently warming the earth further. As the huge C stocks in the soil have the potential to either amplify or attentuate global warming, the impacts of climate change on soil C dynamics require urgent investigation. A more comprehensive representation of ecosytem C exchange is needed, as well as the mechanisms involved, if we want to decrease $CO_2$ efflux from soils and ensure these huge C sinks are stabilised, or potentially even increased such that the biosphere can sequester more atmospheric $CO_2$ and help to stabilise the climate.



**Author contribution:** Marion Nyberg and Mark Hovenden were responsible for conceptualization and project

administration. Marion Nyberg performed data acquisition and curation, formal analysis and writing. Mark
Hovenden developed the methodology and provided supervision.

**Competing interest**

The authors declare that they have no conflict of interest


**Acknowledgements**

We would like to thank Meagan Porter and Rose Brinkhoff for help setting up the experiment and the Tasmanian
Land Conservancy for permission to use the land for the experiment. This research did not receive any specific
grant from funding agencies in the public, commercial, or not-for-profit sectors.

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
