# Peer review of "Warming increases soil respiration in a carbon-rich soil without changing microbial respiratory potential"

_Biogeosciences, 2020_

## Referee Comment (RC1) · Anonymous Referee #1 · 3 Jun 2020

General comments: I enjoyed reading this well written and well presented paper. The use of the orthogonal experimental design is well considered with very thorough statistical analyses. The authors find that warming increases soil respiration but plant community manipulations have no effect on soil respiration. These findings are well interpreted and their implications discussed. I have included some comments below which the authors may wish to consider.

Specific comments: L33-35: Can you briefly expand on, for the reader, why C loss increases with C content please? Is it because there's a higher potential for C loss or a greater proportion of unprotected C or some other mechanism? L120: Please

can you clarify how long the OTC's were in place for and were they in place year-round or during certain seasons only. L146-150: Based on your description of the $CO_2$ flux measurements carried out, it seems that vegetation within the PVC collars was left intact. If this is the case your $CO_2$ flux measurements will have included root, shoot and soil respiration which amounts to ecosystem respiration rather than soil respiration as described. Given that the focus of the paper is on soil respiration I think the contribution of plant respiration to the in situ $CO_2$ flux measurements should be addressed. L150-151: It is not entirely clear what you did here. Did you measure the efflux rate three separate times and take an average of that or did you measure the $CO_2$ concentration at three separate time points and use this to calculate the efflux rate? Please can you clarify this. L157-159: How deep is the organic horizon in these soils? Does 5cm depth cover the whole organic horizon? If not, can you include some details on how representative the top 5 cm of soil might be of the hole of the organic horizon? L421-424: Please can you clarify what you mean by "surface soil layers". I assume from the context you mean soil <5cm deep, but it is not entirely clear from the way it currently reads. Would it be possible to speculate on the variation in temperature between the soil surface and at 5cm depth from literature? This would be useful information to have here, if it exists. L501-502: I think it is worth considering here (or at another appropriate place within this paragraph) that increased C input not only stimulates microbial C mineralisation and C efflux but also increases stable SOM formation through microbial decomposition products. I appreciate that SOM formation is not the focus of this work but I think for balance it is worth highlighting the multiple fates of soil C inputs. L518: In this paragraph you rightly discuss the limitations of your (and most) soil incubations. You mention roots and macrofauna as being absent from the incubated soils. It strikes me that mycorrhizal fungi, which play an important role in soil C dynamics and indeed Rs, are absent from your discussions here. I am not familiar with the plant species at your field sites and they may not be mycorrhizal in which case their omission makes sense, however it the plant communities in question are mycorrhizal it would be worth acknowledging the potential consequences of this in

your incubation experiments.

Technical corrections: L32: Delete .. "and so on".. L44: Correct to: "The effects of temperature on.. " L147: correct "m2" to include the "2" as superscript. The use of subscript rather than superscript occurs a few times throughout (e.g. L223 & 224). This may be a formatting error in the conversion to pdf or the authors personal preference, just check that it is as you want it and aligns with journal specifications. L317: Delete . . ."the situation". . . L338-340: For clarity and flow I suggest re-writing this sentence to: "Post hoc analysis revealed the greatest differences in k were observed between; i) warmed x no removal and warmed x dominant removal plots, and ii) warmed x dominant removal and ambient x dominant removal plots." L401: I think the word "warming" is missing from this sentence. I assume it should read: "There are 4 possible mechanisms whereby warming could have increased Rs: " L532: Consider re-writing to: "Large C stocks within this type of peaty habitat are important for the global C cycle, . . . "

―――――――――――――――――――――

---

## Referee Comment (RC2) · Anonymous Referee #2 · 8 Jun 2020

Please see PDF for comments with correct formatting:

Review of "Warming increases soil respiration in a carbon-rich soil without changing microbial respiratory potential" by Nyberg and Hovenden.

The study presents results from an established warming experiment in carbon-rich soils located in Tasmania. It investigates the mechanisms that drive warming-induced responses in soil respiration, and whether changes in the plant community or changes in microbial soil respiration potential are important drivers. The researchers used both field manipulation and laboratory incubation experiments. There was a consistent effect of warming over time and across all plant community treatments in the field experiment, suggesting that plant community composition manipulations do not influence soil respiration responses to warming. Warming also did not affect microbial respiration in incubation experiments. They conclude that the warming response is most likely due to increased autotrophic respiration and more labile substrate availability to the rhizosphere. Overall the study presents novel results, is well-written and represents obvious effort and contribution to the field. There are few studies that investigate the mechanisms by which soil respiration will increase as a result of warming in carbon-rich soils. Yet, these soils are most likely to contribute to CO2 efflux when warmed. I have a few suggestions for the authors to consider that may enhance their message: • I found it a bit strange that plant community composition data was not presented. In the Discussion (line 475) they state that removing the dominant species did not appear to cause any functional shifts in the community, and that the dominant species may have been replaced by a functionally-similar species. If the authors are able to present community composition data after 1 year, that would help clarify whether the overall plant community changed in some way after removal of the dominant species vs. random removal vs. the control. It would answer the question of whether the plant community was really altered enough to expect possible changes, or whether the plant community treatment just wasn't strong enough to elicit changes. • It's unclear in the Methods whether the removed plant biomass is replaced on the plots as litter or just completely removed. Please clarify this point. • The models presented in Fig 3 and Fig 4 came across as an off-shoot from the main story. If developed further, perhaps including data from similar studies conducted in other soil types, I thought that those two figures could be expanded into a different manuscript that is more broad-reaching. I suggest you remove that information from this paper and just focus in on the experimental results. • I was surprised that there wasn't more of a mention of the effect of warming on soil microbial community composition in the discussion. There is a broad base of literature on this topic, and it is likely that warming/drying not only alters microbial physiology but also community structure. I suggest you expand the background literature and Discussion to address this point. For example: https://doi.org/10.1098/rstb.2019.0112

A few minor edits: Line 11: capitalize Earth Line 17: due to plant community Line 34: it has been suggested Line 53: This would be more effective if you specifically mention Century model examples Line 120, this sentence is unclear: The experiment consists of forty 2 x 2 metre plots, with 3 metres between each plot, of which 20 were warmed using hexagonal polycarbonate 120 open-top chambers (OTC) with an internal diameter of 1.5 m, with the remainder of being unwarmed, ambient plots. Line 123, sentence starting with: "To control for possible effects of removing..." is run-on and difficult to understand. Revise. Table 1: indicate significant differences Line 268-270: "Neither removal, i.e. neither dominant nor random biomass removal ($F_{2,33}$=0.89, P=0.42), nor a warming x removal interaction ($F_{2,33}$=0.57, 270 P=0.57) affected $CO_2$ efflux, as indicated by ANCOVA.", is awkwardly worded. Please revise. Line 401: Change to: There are 4 possible mechanisms whereby Rs could have increased (or similar)

Please also note the supplement to this comment:
https://www.biogeosciences-discuss.net/bg-2020-144/bg-2020-144-RC2-supplement.pdf
* * *

---

## Author Response (AR1)

**Response to reviews of the manuscript:** Warming increases soil respiration in a carbon-rich soil without changing microbial respiratory potential

We thank the reviewers for their detailed comments and have thoroughly revised the manuscript accordingly. Below we provide a detailed response to each point, indicating how and where we have incorporated the suggestions and requests.

**Response to reviewers' comments (responses provided in BLUE)**

**Reviewer 1:**

General comments: I enjoyed reading this well written and well presented paper. The use of the orthogonal experimental design is well considered with very thorough statistical analyses. The authors find that warming increases soil respiration but plant community manipulations have no effect on soil respiration. These findings are well interpreted and their implications discussed. I have included some comments below which the authors may wish to consider.

L33-35: Can you briefly expand on, for the reader, why C loss increases with C content please? Is it because there's a higher potential for C loss or a greater proportion of unprotected C or some other mechanism?

We have revised the text in response to this worthwhile suggestion, expanding on this point (lines 33-35). Specifically, we note that soils with larger C stocks have greater susceptibility to warming, as there is increased substrate for decomposition, with the contention supported by reference to the literature.

L120: Please can you clarify how long the OTC's were in place for and were they in place year- round or during certain seasons only.

This point has been added to the MS (lines 122-123). OTCs have been in place continuously since the experiment was established in 2014 and are currently still operating.

L146-150: Based on your description of the $CO_2$ flux measurements carried out, it seems that vegetation within the PVC collars was left intact. If this is the case your $CO_2$ flux measurements will have included root, shoot and soil respiration which amounts to ecosystem respiration rather than soil respiration as described. Given that the focus of the paper is on soil respiration I think the contribution of plant respiration to the in situ $CO_2$ flux measurements should be addressed.

This is definitely an important suggestion by the reviewer and it is very worthwhile making the situation crystal clear. We have added a more detailed explanation in the methods section to clarify this point (lines 156-161). Further, we addressed this fair point in the discussion section as well (line 413). The reviewer is correct in that our measurements of soil respiration also included a contribution by plant biomass. However, we decided not to completely remove plants from the collars because bare ground is practically non-existent in this ecosystem, partly because of the peaty soil and continuous plant cover. Therefore, we were concerned that removing all plants would create a highly unrepresentative sample of the ecosystem.

We did clip the vegetation within the collars to just above ground height in order to reduce the contribution of aboveground plant parts, but the peaty soils contain a continuous root mat that we were not keen to disturb. Our main concern was that removing all plant biomass would cause the peat to dry out unreasonably quickly, creating cracking and further drying. Therefore, we have maintained our use of the term soil respiration the manuscript, rather than changing to ecosystem respiration, but have pointed out in the methods and discussion that our field measurements include a contribution from roots and a minor contribution from plant shoots.

L150-151: It is not entirely clear what you did here. Did you measure the efflux rate three separate times and take an average of that or did you measure the $CO_2$ concentration at three separate time points and use this to calculate the efflux rate? Please can you clarify this.

The text has been altered to clarify this (lines 162-163). On each occasion, three complete measurements of *in situ* soil respiration, each lasting several minutes, were made in each plot. The results of these three estimations were averaged and used to define the $CO_2$ efflux rate and this single composite value was used in subsequent analyses.

L157-159: How deep is the organic horizon in these soils? Does 5cm depth cover the whole organic horizon? If not, can you include some details on how representative the top 5 cm of soil might be of the hole of the organic horizon?

This has been clarified in the MS (lines 167-168). The organic horizon at the site is deep, up to 1m in depth. However, the 5cm sampling depth is representative of the zone in which most microbial activity occurs in peaty soils (Fisk et al., 2003). Further, the soil profile is fairly consistent for the top ~20 cm and the top 5 cm is indeed representative of this upper layer of soil.

L421-424: Please can you clarify what you mean by "surface soil layers". I assume from the context you mean soil <5cm deep, but it is not entirely clear from the way it currently reads. Would it be possible to speculate on the variation in temperature between the soil surface and at 5cm depth from literature? This would be useful information to have here, if it exists.

This section has been expanded to clarify the situation (lines 437-442). While we do not have measurements of the soil depth-temperature profile, and such measurements appear to be rare in the literature, we added some speculation on the variation in soil temperature along the profile as the reviewer suggested. Specifically, as most soil microbial activity occurs in the uppermost few centimetres, it is possible the most biologically-active soil layer was warmed more than the amount measured, partially accounting for the large increase in Rs. However, the warming-depth profile at Silver Plains is unknown and also largely unreported from other warming experiments, except at greater depths (e.g. 0-5 cm versus 5-15 cm (Hollister et al., 2006)).

L501-502: I think it is worth considering here (or at another appropriate place within this paragraph) that increased C input not only stimulates microbial C mineralisation and C efflux but also increases stable SOM formation through microbial decomposition products. I appreciate that SOM formation is not the focus of this work but I think for balance it is worth highlighting the multiple fates of soil C inputs.

This is a valid point and the MS has been altered to include this (lines 543-545). Specifically, we now state that an increase in the input of easily degradable C would both promote microbial activity (Wan et al., 2005; Hogberg and Read, 2006), potentially stimulating soil C efflux, as well as increase formation of stable SOM through microbial decomposition products (Sokol et al., 2019).

L518: In this paragraph you rightly discuss the limitations of your (and most) soil incubations. You mention roots and macrofauna as being absent from the incubated soils. It strikes me that mycorrhizal fungi, which play an important role in soil C dynamics and indeed Rs, are absent from your discussions here. I am not familiar with the plant species at your field sites and they may not be mycorrhizal in which case their omission makes sense, however it the plant communities in question are mycorrhizal it would be worth acknowledging the potential consequences of this in your incubation experiments.

This is a very valid point and we have added specific statements regarding the importance of mycorrhizal fungi to SOM formation as well as how removing their influence is likely to have altered our results (lines 568-659).

*Technical corrections:*

L32: Delete .. "and so on"..

Text altered as suggested

L44: Correct to: "The effects of temperature on.. "

Text altered as suggested

L147: correct "m2" to include the "2" as superscript. The use of sub- script rather than superscript occurs a few times throughout (e.g. L223 & 224). This may be a formatting error in the conversion to pdf or the authors personal preference, just check that it is as you want it and aligns with journal specifications.

Text altered as suggested

L317: Delete ..."the situation"...

Text altered as suggested

L338-340: For clarity and flow I suggest re-writing this sentence to: "Post hoc analysis revealed the greatest differences in k were observed between; i) warmed x no removal and warmed x dominant removal plots, and ii) warmed x dominant removal and ambient x dominant removal plots."

Text altered as suggested

L401: I think the word "warming" is missing from this sentence. I assume it should read: "There are 4 possible mechanisms whereby warming could have increased Rs: "

Text altered as suggested

L532: Consider re-writing to: "Large C stocks within this type of peaty habitat are important for the global C cycle, ..."

Text altered as suggested

**Reviewer 2:**

The study presents results from an established warming experiment in carbon-rich soils located in Tasmania. It investigates the mechanisms that drive warming-induced responses in soil respiration, and whether changes in the plant community or changes in microbial soil respiration potential are important drivers. The researchers used both field manipulation and laboratory incubation experiments. There was a consistent effect of warming over time and across all plant community treatments in the field experiment, suggesting that plant community composition manipulations do not influence soil respiration responses to warming. Warming also did not affect microbial respiration in incubation experiments. They conclude that the warming response is most likely due to increased autotrophic respiration and more labile substrate availability to the rhizosphere.

Overall the study presents novel results, is well-written and represents obvious effort and contribution to the field. There are few studies that investigate the mechanisms by which soil respiration will increase as a result of warming in carbon-rich soils. Yet, these soils are most likely to contribute to CO2 efflux when warmed. I have a few suggestions for the authors to consider that may enhance their message:

I found it a bit strange that plant community composition data was not presented. In the Discussion (line 475) they state that removing the dominant species did not appear to cause any functional shifts in the community, and that the dominant species may have been replaced by a functionally-similar species. If the authors are able to present community composition data after 1 year, that would help clarify whether the overall plant community changed in some way after removal of the dominant species vs. random removal vs. the control. It would answer the question of whether the plant community was really altered enough to expect possible changes, or whether the plant community treatment just wasn't strong enough to elicit changes.

This is a valid point. We have included a supplementary figure (Fig. S1) showing plant community composition responses to the treatments at the end of the first growing season after treatments began. In addition, we have also added a relevant section to the discussion (lines 510-519). However, we do not present detailed plant community composition data as a full analysis is beyond the scope of this manuscript and will be published elsewhere.

It's unclear in the Methods whether the removed plant biomass is replaced on the plots as litter or just completely removed. Please clarify this point.

Text altered as suggested to clarify that removed plant biomass was completely removed from plots and not replaced as litter (lines 133-134).

The models presented in Fig 3 and Fig 4 came across as an off-shoot from the main story. If developed further, perhaps including data from similar studies conducted in other soil types, I thought that those two figures could be expanded into a different manuscript that is more broad-reaching. I suggest you remove that information from this paper and just focus in on the experimental results.

It would be interesting indeed to create a separate manuscript as the reviewer suggests but, as they acknowledge, this would require more results from different sites. We will definitely keep act upon this suggestion. However, we believe that the data presented in Figs 3 and 4 is key to understanding the field observations because they help to demonstrate the patterns we discovered more clearly. These figures allow readers to easily identify the relative influences of soil temperature and SWC on C efflux rate and the specific conditions that led to the largest influence of experimental warming on C efflux. Therefore, while we do appreciate the reviewers point and helpfulness, we feel that these figures make an important contribution to this manuscript and would prefer to include them here.

I was surprised that there wasn't more of a mention of the effect of warming on soil microbial community composition in the discussion. There is a broad base of literature on this topic, and it is likely that warming/drying not only alters microbial physiology but also community structure. I suggest you expand the background literature and Discussion to address this point. For example: **https://doi.org/10.1098/rstb.2019.0112**

This is a valid point, and a new section '4.1.2 Alteration of microbial community composition and function', has been added to expand on this (line 476).

Line 11: capitalize Earth

Text altered as suggested

Line 17: due to plant community

Text altered as suggested

Line 34: it has been suggested

Text altered for clarity

Line 53: This would be more effective if you specifically mention Century model examples

This is a valid point and a CENTURY model example has been included in the text (lines 50-53).

Line 120, this sentence is unclear: The experiment consists of forty 2 x 2 metre plots, with 3 metres between each plot, of which 20 were warmed using hexagonal polycarbonate 120 open-top chambers (OTC) with an internal diameter of 1.5 m, with the remainder of being unwarmed, ambient plots.

Text altered for clarity (lines 123-126): "The experiment consists of forty 2 x 2 metre plots, with 3 metres between each plot. 20 of the plots were warmed year-round using hexagonal polycarbonate open-top chambers (OTC) with an internal diameter of 1.5 m, and the remainder were unwarmed, ambient plots."

Line 123, sentence starting with: "To control for possible effects of removing..." is run-on and difficult to understand. Revise.

We have revised the text to make two shorter sentences. The text now reads (lines 129-133): "To control for possible effects of removing biomass during the dominant species removal treatment, we removed biomass from one additional warmed and unwarmed plot in every second block. We removed the same amount of biomass as from the "dominant removal" plots in the same block, however, biomass was removed randomly from across the plot, rather than from a single species (henceforth termed "random removal" plots)."

Table 1: indicate significant differences

Significant differences within months between warming treatments have been indicated and the table heading updated to explain the addition.

Line 268-270: "Neither removal, i.e. neither dominant nor random biomass removal ($F_{2,33}=0.89$, P=0.42), nor a warming x removal interaction ($F_{2,33}=0.57$, 270 P=0.57) affected CO2 efflux, as indicated by ANCOVA.", is awkwardly worded. Please revise.

This sentence (lines 283-285) has been revised to now read: "Neither removal treatment, ($F_{2,33}=0.89$, P=0.42), nor a warming x removal interaction ($F_{2,33}=0.57$, P=0.57) affected $CO_2$ efflux, as indicated by ANCOVA."

Line 401: Change to: There are 4 possible mechanisms whereby Rs could have increased (or similar)

Text altered as suggested (lines 416-417)

**Warming increases soil respiration in a carbon-rich soil without changing microbial respiratory potential**

Marion Nyberg[1], Mark J. Hovenden[1]

[1]School of Natural Sciences, University of Tasmania, Hobart, 7001, Australia

5 *Correspondence to:* Marion Nyberg (current affiliation University of British Columbia) (mnybe1@mail.ubc.ca)

**Abstract.** Increases in global temperatures due to climate change threaten to tip the balance between carbon (C) fluxes, liberating large amounts of C from soils. Evidence of warming-induced increases in $CO_2$ efflux from soils has led to suggestions that this response of soil respiration ($R_S$) will trigger a positive land C–climate feedback

10 cycle, ultimately warming the Earth further. Currently, there is little consensus about the mechanisms driving the warming-induced $R_S$ response, and there are relatively few studies from ecosystems with large soil C stores. Here, we investigate the impacts of experimental warming on $R_S$ in the C-rich soils of a Tasmanian grassy sedgeland, and whether alterations of plant community composition or differences in microbial respiratory potential could contribute to any effects. *In situ*, warming increased $R_S$ on average by 28% and this effect was consistent over

15 time and across plant community composition treatments. In contrast, warming had no impact on microbial respiration in incubation experiments. Plant community composition manipulations did not influence $R_S$ or the $R_S$ response to warming. Processes driving the $R_S$ response in this experiment were, therefore, not due to plant community effects and are more likely due to increases in belowground autotrophic respiration and the supply of labile substrate through rhizodeposition and root exudates. $CO_2$ efflux from this high-C soil increased by more

20 than a quarter in response to warming, suggesting inputs need to increase by at least this amount if soil C stocks are to be maintained. These results indicate the need for comprehensive investigations of both C inputs and losses from C-rich soils if efforts to model net ecosystem C exchange of these crucial, C-dense systems are to be successful.

25 ## 1 Introduction

Globally, more carbon (C) is stored in soils than the amount of C in the atmosphere and in plants combined (Canadell et al., 2007). Simple physiology suggests that soil respiration ($R_S$) rates will increase as soil temperatures rise (Gillooly et al., 2001), stimulating $CO_2$ emissions from the soil – a response that has the potential to outweigh plant productivity responses to global warming and lead to a net loss of C from soils (Melillo et al.,

30 2017). Recently, numerous studies have suggested that global warming is indeed disturbing the balance between ecosystem C inputs and outputs (Melillo et al., 2017). This suggests the possibility of a positive feedback whereby warming increases C efflux from soils, which accelerates climate change leading to further C losses. (Bridgham et al., 2008; Melillo et al., 2017; Bond-Lamberty et al., 2018). Importantly, it is possible that warming-induced C losses increase with soil C content, as soils with large C stocks have greater susceptibility to warming since there

35 is more substrate available for decomposition, and therefore soils storing the most C could shift from C sinks to C sources (Crowther et al., 2016).

**Commented [MN1]: R1**
"Delete .. "and so on".."
Text altered as suggested

**Commented [MN2]: R1:** "Can you briefly expand on, for the reader, why C loss increases with C content please? Is it because there's a higher potential for C loss or a greater proportion of unprotected C or some other mechanism?"
We have revised the text in response to this worthwhile suggestion, expanding on this point (lines 33-35). Specifically, we note that soils with larger C stocks have greater susceptibility to warming, as there is increased substrate for decomposition, with the contention supported by reference to the literature.

[revised manuscript text omitted]

**Commented [MN5]: R1**
"Please can you clarify how long the OTC's were in place for and were they in place year- round or during certain seasons only."
This point has been added to the MS (lines 122-123). OTCs have been in place continuously since the experiment was established in 2014 and are currently still operating.

**Commented [MN6]: R2**
"this sentence is unclear: The experiment consists of forty 2 x 2 metre plots, with 3 metres between each plot, of which 20 were warmed using hexagonal polycarbonate 120 open-top chambers (OTC) with an internal diameter of 1.5 m, with the remainder of being unwarmed, ambient plots."
Text altered for clarity (lines 123): "The experiment consists of forty 2 x 2 metre plots, with 3 metres between each plot. 20 of the plots were warmed year-round using hexagonal polycarbonate open-top chambers (OTC) with an internal diameter of 1.5 m, and the remainder were unwarmed, ambient plots."

**Commented [MN7]: R2**
"It's unclear in the Methods whether the removed plant biomass is replaced on the plots as litter or just completely removed. Please clarify this point."
Text altered as suggested to clarify that removed plant biomass was completely removed from plots and not replaced as litter (lines 133-134).

[Figure]

**Figure 1. Conceptual diagram for the experimental design of the Silver Plains warming experiment. Each block contains a warmed and unwarmed plot with no species removed (WNR) and (ANR) respectively; a warmed and unwarmed plot with the dominant species removed (WDR) and (ADR) respectively; and in every second block, i.e. in four blocks, there is a warmed and unwarmed plot with random biomass removal (WRR) and (ARR) respectively.**

Air temperature at 5 cm height and soil temperature at 5 cm depth in each plot was logged continuously with iButton dataloggers. Over the entire five year period, the warming treatment increased air temperature 5 cm above the soil surface by 1.56°C ($P<0.004$) and soil temperature at 5 cm depth by 1.29°C ($P<0.001$).

**2.3 *In situ* methods**

A 50 mm length of 100 mm diameter PVC pipe was inserted into the soil to a depth of 2 cm, extending 3 cm above ground height, within the centre 0.25 m² of each plot for soil respiration measurements. Soil respiration was measured with a $CO_2/H_2O$ infrared gas analyser (IRGA) (Licor, model LI-6400) with attachment of a Licor 6400-09 soil chamber, which attached to PVC collars. Bare ground is practically non-existent in this ecosystem and the soil is extremely peaty. Therefore, vegetation in collars was not removed but was regularly clipped to just above ground height to minimise the influence of aboveground plant respiration. Our measurement of soil respiration thus did include small contributions of shoot respiration, but as soil at the site is peaty with extensive horizontal root growth, any respiration measurement from this site would include a substantial amount of respiration from belowground plant biomass. Respiration was measured *in situ* monthly from August 2017 to June 2018. On each occasion, three complete measurements of *in situ* soil respiration in each plot were averaged and used to define the $CO_2$ efflux rate. The average value of these three measurements was used in subsequent analyses. Soil temperature and moisture in each plot were measured at the exact same time as the soil respiration

**Commented [MN8]: R1**
"Based on your description of the CO2 flux measurements carried out, it seems that vegetation within the PVC collars was left intact. If this is the case your CO2 flux measurements will have included root, shoot and soil respiration which amounts to ecosystem respiration rather than soil respiration as described. Given that the focus of the paper is on soil respiration I think the contribution of plant respiration to the in situ CO2 flux measurements should be addressed."
This is definitely an important suggestion by the reviewer and it is very worthwhile making the situation crystal clear. We have added a more detailed explanation in the methods section to clarify this point (lines 156-161). Further, we addressed this fair point in the discussion section as well (line 413). The reviewer is correct in that our measurements of soil respiration also included a contribution by plant biomass. However, we decided not to completely remove plants from the collars because bare ground is practically non-existent in this ecosystem, partly because of the peaty soil and continuous plant cover. Therefore, we were concerned that removing all plants would create a highly unrepresentative sample of the ecosystem. We did clip the vegetation within the collars to just above ground height in order to reduce the contribution of aboveground plant parts, but the peaty soils contain a continuous root mat that we were not keen to disturb. Our main concern was that removing all plant biomass would cause the peat to dry out unreasonably quickly, creating cracking and further drying. Therefore, we have maintained our use of the term soil respiration the manuscript, rather than changing to ecosystem respiration, but have pointed out in the methods and discussion that our field measurements include a contribution from roots and a minor contribution from plant shoots.

**Commented [MN9]: R1**
"It is not entirely clear what you did here. Did you measure the efflux rate three separate times and take an average of that or did you measure the CO2 concentration at three separate time points and use this to calculate the efflux rate? Please can you clarify this."
The text has been altered to clarify this (lines 162-163). On each occasion, three complete measurements of *in situ* soil respiration, each lasting several minutes, were made in each plot. The results of these three estimations were averaged and used to define the $CO_2$ efflux rate and this single composite value was used in subsequent analyses.

measurements on each occasion. Soil temperature was measured with a soil thermocouple probe (LiCor 6000-09TC) attached to the LI-6400. Volumetric soil water content (SWC) was estimated at 5 locations in each plot using a hand held TDR probe at 0-5cm depth. Although the organic horizon in this soil is up to 1m in depth, the 5cm sampling depth is representative for the zone in which most microbial activity occurs in peaty soils (Fisk et al., 2003). The 5 separate measurements of SWC were then averaged to obtain one SWC value per plot on each measuring occasion.

Six randomly placed soil samples, amounting to a total of approximately 25-30 g fresh weight, were collected from each plot using a 1.5 cm diameter hand corer to a depth of 5 cm below ground level, twice throughout the year. Samples were collected on the 02/03/18, representing the end of summer, or growing season soil, and on the 25/06/18, representing winter soils.

**2.4 Laboratory incubations**

Soil cores collected *in situ* were immediately placed on ice for return to the laboratory, where they were refrigerated (4°C) overnight. The following day, the samples were composited at the plot level and sieved through a 4 mm sieve for one minute to remove leaves and large roots. A 10 g fresh-weight sub-sample was removed and oven dried from each composite sample for the determination of total soil C. Each subsample was ground to a powder in a Retsch Mixer Mill (MM200, Retsch GmbH, Haan) and then C content was analysed by combustion in a Perkin Elmer 2400 Series II Elemental Analyser (Perkin Elmer Australia, Melbourne). The remaining soil was used immediately for laboratory incubations to determine microbial respiration, as detailed below.

Microbial respiration as a function of temperature was determined by incubation using soils sampled in the Silver Plains warming experiment at the end of summer and in mid-winter 2018. For each plot, three replicate samples weighing four to eight grams from the composite sample were placed in 100 mL specimen jars, each of which was incubated at a different temperature. Each sample was wetted to bring them to 90% of field capacity for winter soils and 60% of field capacity for summer soils to represent prevailing soil moisture conditions in each respective season. Once water was added to all soil samples, specimen jars were placed in 500 ml preserving jars with tightly fitting lids containing a septum to allow gas headspace samples to be collected by syringe. Jars were stored in dark incubation cabinets at temperatures at one of 10, 17 or 25°C, with one sample from each plot at each temperature. Headspace gas of jars were sampled (20 ml) using a syringe on days 1, 2, 4, 5, 7, 9, 12, 15, 19, 23, 29, 35, 49, 56, 63. After extracting samples from each jar, headspace samples were analysed for $CO_2$ concentration, representing soil respiration, and microbial respiratory potential was thus defined as the rate of $CO_2$ release. To analyse headspace gas, samples were injected directly into an infrared gas analyser (LI-6262, Li-Cor, Lincoln, NE). After measurements were taken and analysed, jars were ventilated for 20 minutes and headspace gas equilibrated with atmospheric air. Following this, lids were replaced and headspace gas was sampled and analysed again to obtain starting $CO_2$ concentration for each jar. C mineralisation over the sample period was calculated from the increase in headspace $CO_2$ concentration.

Total C mineralisation over the entire incubation period was simply the sum of the amount of C mineralised over each sample period. Daily C mineralization results (dC/dt) were analysed using non-linear curve fitting routines

**Commented [MN10]:** R1

"How deep is the organic horizon in these soils? Does 5cm depth cover the whole organic horizon? If not, can you include some details on how representative the top 5 cm of soil might be of the hole of the organic horizon?"

The text has been altered to clarify this (lines 162-163). On each occasion, three complete measurements of *in situ* soil respiration, each lasting several minutes, were made in each plot. The results of these three estimations were averaged and used to define the $CO_2$ efflux rate and this single composite value was used in subsequent analyses.

[revised manuscript text omitted]

**Commented [MN12]:** R2
""Neither removal, i.e. neither dominant nor random biomass removal (F2,33=0.89, P=0.42), nor a warming x removal interaction (F2,33=0.57, 270 P=0.57) affected CO2 efflux, as indicated by ANCOVA.", is awkwardly worded. Please revise."
This sentence has been revised to now read: "Neither removal treatment, ($F_{2,33}$=0.89, P=0.42), nor a warming x removal interaction ($F_{2,33}$=0.57, P=0.57) affected $CO_2$ efflux, as indicated by ANCOVA."

**Commented [MN13]:** R2
"The models presented in Fig 3 and Fig 4 came across as an off-shoot from the main story. If developed further, perhaps including data from similar studies conducted in other soil types, I thought that those two figures could be expanded into a different manuscript that is more broad-reaching. I suggest you remove that information from this paper and just focus in on the experimental results."
It would be interesting indeed to create a separate manuscript as the reviewer suggests but, as they acknowledge, this would require more results from different sites. We will definitely keep act upon this suggestion. However, we believe that the data presented in Figs 3 and 4 is key to understanding the field observations because they help to demonstrate the patterns we discovered more clearly. These figures allow readers to easily identify the relative influences of soil temperature and SWC on C efflux rate and the specific conditions that led to the largest influence of experimental warming on C efflux. Therefore, while we do appreciate the reviewers point and helpfulness, we feel that these figures make an important contribution to this manuscript and would prefer to include them here.

[revised manuscript text omitted]

**Commented [MN15]: R1**
Further clarification of our use of soil respiration as opposed to ecosystem respiration

**Commented [MN16]: R2**
"Change to: There are 4 possible mechanisms whereby Rs could have increased (or similar)"
Text altered as suggested

observed effect of soil water content and soil temperature on $R_S$ was anticipated, however the degree to which warming enhanced the response of $R_S$ to temperature, was greater than expected. This observation could be explained by the greater effect of warming on air temperature than soil temperature at 5cm, thus considering most soil microbial activity occurs in the uppermost few centimetres, it is possible the most biologically-active soil layer was warmed more than the amount measured, partially accounting for the large increase in $R_S$. However, the warming-depth profile at Silver Plains is unknown and also largely unreported from other warming experiments, except at greater depths (e.g. 0-5 cm versus 5-15 cm (Hollister et al., 2006)). Clearly, the influence of warming treatments on the soil temperature-depth profile is an area that requires further investigation. Interestingly, our results show that the degree of stimulation by warming increased as soil temperature increased, i.e., there was a greater warming-induced stimulation of C efflux when soil was warm than when it was cold (Fig. 2). This contrasts previous findings that indicate a greater warming effect on $R_S$ at lower temperatures (Wang et al., 2014).

The large apparent increase in $R_S$ observed *in situ* implied that warming possibly triggered an increase in microbial respiratory potential. Partitioning of $R_S$ in incubation experiments allows the response of microbial respiration to warming to be observed under optimal conditions, controlling for soil water content and soil temperature. In stark contrast to the *in situ* observations, soil incubations revealed no differences in the temperature sensitivity of microbial respiration between warmed and ambient soil. There were no differences among treatments in the total C mineralisation rate measured under laboratory conditions, which would indicate that the ability of the soil microbial community to mineralise soil C was unchanged. This lack of any treatment effect was similar in winter and summer-collected soils, even though there was a strong seasonal effect on the $CO_2$ respiration rate in incubated soils. Winter soils emitted significantly less $CO_2$ than soils collected at the end of summer, a response attributed to decreased access to substrate as an effect of limited enzyme activity in cold temperatures (Suseela et al., 2012). Additionally, despite claims that the warming-induced increase in $R_S$ is due to a strong, positive relationship between the average turnover time of labile C pools and mean annual air temperature (Trumbore et al., 1996), there was no difference in the size of the respired labile C pool ($C_a$) between warmed and ambient soils, or between seasons. Therefore, this suggests that the warming treatment did not increase the temperature sensitivity of labile C decomposition. Additionally, there were no warming or removal treatment effects on the decay constant of the stable C pool ($Y_0$), calculated from $CO_2$ emission rates late in the incubation period. This implies that stable C, which is chemically and physically protected (Schlesinger, 1997), was not sensitive to warming, a response that contrasts results obtained elsewhere (Leifeld and Fuhrer, 2005; Hartley and Ineson, 2008). Thus, lack of a warming effect indicates that warming-induced increases in labile, or stable C temperature sensitivity are not driving the $R_S$ response to warming observed *in situ*.

Essentially, the incubation studies revealed that four years of experimental warming had not altered either the potential for microbial respiration or its inherent temperature response, as soils incubated at the same temperature respired more or less at the same rate, regardless of whether they were collected from warmed or ambient plots. These results indicate that the warming-induced stimulation of $R_S$ *in situ* was not due to changes in the inherent temperature response of microbial respiration. Considering the soil incubation experiment decoupled microbial respiration from plant activity and particularly C inputs, the lack of a warming treatment effect on C emissions in the incubation experiments is evidence that plants play a large role in the respiration response.

**Commented [MN17]: R1**
"Please can you clarify what you mean by "surface soil layers". I assume from the context you mean soil <5cm deep, but it is not entirely clear from the way it currently reads. Would it be possible to speculate on the variation in temperature between the soil surface and at 5cm depth from literature? This would be useful information to have here, if it exists."

This section has been expanded to clarify the situation (lines 437-442). While we do not have measurements of the soil depth-temperature profile, and such measurements appear to be rare in the literature, we added some speculation on the variation in soil temperature along the profile as the reviewer suggested. Specifically, as most soil microbial activity occurs in the uppermost few centimetres, it is possible the most biologically-active soil layer was warmed more than the amount measured, partially accounting for the large increase in $R_S$. However, the warming-depth profile at Silver Plains is unknown and also largely unreported from other warming experiments, except at greater depths (e.g. 0-5 cm versus 5-15 cm (Hollister et al., 2006)).

**4.1.2 Alteration of microbial community composition and function**

The role of microbial community composition and function in the respiration response to warming is complex as it encompasses multiple possible factors that could lead to changes in respiration rates (Bargett & Caruso 2020; Karhu et al., 2014). These factors include changes in individual microbial physiology, whereby temperature affects the rate at which microbes can take up and metabolise substrate (Hopkins et al., 2014), genetic changes within species, indicating possible adaption to specific environmental conditions (Karhu et al., 2014), competition between species (Sheik et al., 2011), and changes in community composition to support taxa that thrive in warmer or drought prone conditions (Bardgett & Caruso 2020) . From the $R_S$ response observed *in situ*, a shift in microbial community composition and function seems to be a plausible driving factor. However, laboratory incubations of soil indicated the temperature response of respiratory potential in this study did not differ due to an increase in temperature in the warming treatment. If there were changes to the microbial community, they did not appear to have a role in altering the response of respiration to warming, implying the $R_S$ response was not due to a shift in microbial community and function.

490

**4.1.3 Influence through change in plant species community composition**

Plant community compositional change drives ecosystem responses to global changes, particularly when it involves shifts in the dominance or abundance of plant functional types (Bret-Harte et al., 2008). This is particularly true with $R_S$ and global warming, as warming-related changes in plant functional types, and hence the resources they input to the soil, are highly likely to occur (Saleska et al., 2002). By investigating the effects of dominant species removal, random biomass removal, and warming on the response of respiration, there is scope to gain insight into future ecosystem dynamics under a changing climate. Removal of a dominant species from an ecosystem has promoted species diversity and altered ecosystem function, implying dominants reduce the establishment of other species (Wardle et al., 1999). Metcalfe et al., (2011) highlight the significant role that functional traits of the dominant species hold on many soil processes, including decomposition and respiration and hence one would expect to observe these effects in this study. Plants modify local soil conditions through root exudations of hormones, sugars, phenolics and amino acids, essentially structuring the rhizosphere microbial community composition. This means that changes in plant community composition have the potential to affect $R_S$ and thus ecosystem functioning (Van Nuland et al., 2016), and hence the motive to investigate how $R_S$ responds to the combination of warming and manipulated plant community composition. Results from Silver Plains demonstrate neither removal of the dominant plant species nor random removal of biomass had any effect on $R_S$ or the temperature response of $R_S$ *in situ*, and very little effect on microbial respiratory potential. This suggests that, in line with previous studies, temperature had greater control on $R_S$ than variation in plant community composition (Duval and Radu, 2018).

Multi-dimensional scaling analysis of plant community composition indicated that the removal of the dominant species did tend to shift community composition, but this change was not substantial in comparison to the natural

Commented [MN18]: R2
"I was surprised that there wasn't more of a mention of the effect of warming on soil microbial community composition in the discussion. There is a broad base of literature on this topic, and it is likely that warming/drying not only alters microbial physiology but also community structure. I suggest you expand the background literature and Discussion to address this point. For example: https://doi.org/10.1098/rstb.2019.0112"
This is a valid point, and a new section '4.1.2 Alteration of microbial community composition and function', has been added to expand on this (line 476).

variation in community composition within the control plots (Fig. S1). The plant community composition in plots subjected to random biomass removal was similar to that of control plots. Furthermore, the removal of the dominant species did not appear to cause any functional shift within this species-rich community, suggesting it may have been replaced by a functionally-similar species, or there was a compensatory response by functionally-different species (Bret-Harte et al., 2008). Thus, the removal treatments had relatively modest influences on the plant community composition and function, potentially explaining why $R_s$ was similarly unresponsive to the treatments. Nevertheless, the plant community composition did differ considerably among plots across the experiment (Fig. S1). Differences in plant chemistry, morphology and physiology affect the quantity and quality of root and leaf litter, leading to changes in SOM decomposition rates, shifts in microbial respiratory potential and community structure (Van Nuland et al., 2016). However, the consistent response of $R_s$ *in situ* to the warming treatment indicates that warming effects were similar across the variety of plant community composition within this ecosystem.

Despite the absence of an effect on $R_s$ *in situ* from manipulating species composition, microbial respiration dynamics indicated that warming and biomass removal (both random and dominant removal treatments) reduced the intrinsic decay constant of the labile C pool ($k$) in summer soil. Interestingly, biomass removal had the opposite effect in ambient plots. In a previous clipping experiment, which is representative of biomass removal, a decrease in $R_s$ due to clipping was explained as relocation of assimilates to shoots, reduction in the supply of photosynthates to roots, and thus decreased root respiration (Zhou et al., 2010). Hence there is likely to be less available substrate under warming and biomass removal scenarios, and $k$ is therefore lower. Considering this, the interactive effect of warming and biomass removal on $k$ is complex and requires further investigation to explore the mechanistic basis behind the response. The absence of an influence on $R_s$ through variation in plant community composition suggests this mechanism is not driving the warming-induced $R_s$ response to warming.

**4.1.4 Enhanced substrate supply**

**SOM**

As SOM forms mainly from plant litter, warming related increases in both above and belowground primary productivity suggest supply of SOM will be greater under warming (Rustad et al., 2001; Lin et al., 2010; Wu et al., 2011), at least in systems that are not water limited. Additionally, experimental warming often increases leaf-drop, root-turnover and the subsequent decomposition of leaf and root litter (Lu et al., 2013), with the combined effects of warming and higher C inputs on respiration rates reported to be greater than the impact of either factor in isolation (Hopkins et al., 2014). Root and leaf litter have fast turnover times, implying they represent a major source of C for microbial decomposition. Therefore, an increase in the input of easily degradable C would both promote microbial activity (Wan et al., 2005; Hogberg and Read, 2006), potentially stimulating soil C efflux, as well as increasing formation of stable SOM through microbial decomposition products (Sokol et al., 2019). Considering this, an increase in substrate supply seems like a conceivable explanation for the increase in $R_s$ observed *in situ*. However, incubation experiments indicated no influence of warming on the total amount of C between warmed and ambient plots, or on the size of the labile pool ($C_a$), or total C respired. This indicates that substrate supply and availability from plant biomass is similar in warmed and ambient plots. Previous
* * *
**Commented [MN19]: R2**
"I found it a bit strange that plant community composition data was not presented. In the Discussion (line 475) they state that removing the dominant species did not appear to cause any functional shifts in the community, and that the dominant species may have been replaced by a functionally-similar species. If the authors are able to present community composition data after 1 year, that would help clarify whether the overall plant community changed in some way after removal of the dominant species vs. random removal vs. the control. It would answer the question of whether the plant community was really altered enough to expect possible changes, or whether the plant community treatment just wasn't strong enough to elicit changes."
This is a valid point. We have included a supplementary figure (Fig. S1) showing plant community composition responses to the treatments at the end of the first growing season after treatments began. In addition, we have also added a relevant section to the discussion (lines 510-519). However, we do not present detailed plant community composition data as a full analysis is beyond the scope of this manuscript and will be published elsewhere.

**Commented [MN20]: R1**
"I think it is worth considering here (or at another appropriate place within this paragraph) that increased C input not only stimulates microbial C mineralisation and C efflux but also increases stable SOM formation through microbial decomposition products. I appreciate that SOM formation is not the focus of this work but I think for balance it is worth highlighting the multiple fates of soil C inputs."
This is a valid point and the MS has been altered to include this (lines 543-545). Specifically, we now state that an increase in the input of easily degradable C would both promote microbial activity (Wan et al., 2005; Hogberg and Read, 2006), potentially stimulating soil C efflux, as well as increase formation of stable SOM through microbial decomposition products (Sokol et al., 2019).

investigations suggest that despite warming-related increases in litter quantity, enhanced respiration due to increased labile C concentration in soils is likely to offset additional C inputs (Lu et al., 2013), meaning changes to both inputs and losses of soil C could balance each other. Interestingly neither *in situ* $R_S$, nor total soil C or C$a$ was affected by plant community composition manipulations, suggesting substrate supply and availability was similar regardless of warming and removal treatments. This result contrasts with those from previous clipping experiments that demonstrated that biomass removal limits substrate supply (Wan and Luo, 2003; Xue et al., 2016). Overall this suggests that increased substrate supply through SOM is not a driving mechanism behind the warming-induced increase in $R_S$ observed *in situ* although specific tests of this mechanism, such as through the use of stable isotope tracing, would be required to be confident.

**4.1.5 Plant induced alteration to soil microhabitat**

The final mechanism that could be driving the warming-induced increases in $R_S$ are plant induced alterations to the soil microhabitat. In this study, as in most, soil for incubations was not analysed as intact soil cores, rather being sieved and homogenised, altering the microhabitat conditions. This is potentially problematic, as it is assuming that rhizosphere processes, including contributions from mycorrhizal fungi are not influencing the overall $R_S$ response. Previous studies have demonstrated the important role roots play in stabilising SOM (Hinsinger et al., 2009), with disturbed soils having a lower capacity to protect SOM due to mechanical disruption of macroaggregates, and hence C is more readily decomposed by microorganisms (Beare et al., 1994). Many, and potentially all, of the plant species at the site have associations with arbuscular mycorrhizae, which are known to increase SOM formation both directly as well as through their influence on soil aggregation (Rillig et al., 2001). Thus, our incubations would have removed this important contribution, reducing SOM formation and potentially increasing C mineralisation rates. Additionally, macrofauna such as earthworms and nematodes play an important role in the early stages of SOM decomposition (Wardle et al., 2004), and therefore the absence of these species from the incubations could also have influenced the rates of C efflux. Considering this, it is possible that through homogenisation of soil in incubation studies, soil C dynamics and decomposition rates are confounded by disturbances to the soil microhabitat.

**4.2 Implications**

Large C stocks within this type of peaty habitat are important for the global C cycle, thus understanding potential losses are immensely important for the global C budget. Previous studies on the response of $R_S$ to warming have been largely centred around northern hemisphere sites, and with that there has been large unexplained variability in the response (van Gestel et al., 2018). This implies that the mechanisms behind the $R_S$ response to warming are poorly characterised. Our results indicate that warming-related increases in $CO_2$ efflux from C-rich soils in grassy peatlands are expected in the future, however microbial respiratory potential is not the driving factor, and thus there is a strong link to plant activity and C inputs. Moreover, the results indicate that the impact of warming on soil $CO_2$ efflux is strongly dependent upon both soil temperature and moisture conditions, improving the

**Commented [MN21]: R1**
"In this paragraph you rightly discuss the limitations of your (and most) soil incubations. You mention roots and macrofauna as being absent from the incubated soils. It strikes me that mycorrhizal fungi, which play an important role in soil C dynamics and indeed Rs, are absent from your discussions here. I am not familiar with the plant species at your field sites and they may not be mycorrhizal in which case their omission makes sense, however it the plant communities in question are mycorrhizal it would be worth acknowledging the potential consequences of this in your incubation experiments."
This is a very valid point and we have added specific statements regarding the importance of mycorrhizal fungi to SOM formation as well as how removing their influence is likely to have altered our results (lines 568-659).

**Commented [MN22]: R1**

[revised manuscript text omitted]